# Sea Ice dynamics in the Bransfield Strait, Antarctic Peninsula, during the past 240 years: a multi-proxy intercomparison study

Maria-Elena Vorrath[1], Juliane Müller[1,2,3], Lorena Rebolledo[4,5], Paola Cárdenas[4], Xiaoxu Shi[1],

Oliver Esper[1], Thomas Opel[1], Walter Geibert[1], Práxedes Muñoz[6], Christian Haas[1], Gerhard

Kuhn[1], Carina B. Lange[4,7,8,9], Gerrit Lohmann[1], Gesine Mollenhauer[1,2]

[1]Alfred Wegener Institute, Helmholtz Centre for Polar and Marine Research, Bremerhaven, Germany

[2]MARUM – Center for Marine Environmental Sciences, University of Bremen, Germany

[3]Department of Geosciences, University of Bremen, Germany

[4]Centro de Investigación Dinámica de Ecosistemas Marinos de Altas Latitudes (IDEAL), Universidad Austral de

Chile, Valdivia, Chile

[5]Instítuto Antártico Chileno (INACH), Punta Arenas, Chile

[6]Facultad de Ciencias del Mar, Universidad Católica del Norte, Coquimbo, Chile

[7]Centro Oceanográfico COPAS Sur-Austral, Universidad de Concepción, Chile

[8]Departamento de Oceanografía, Universidad de Concepción, Chile

[9]Scripps Institution of Oceanography, La Jolla, CA 92037, USA

*Correspondence to*: Maria-Elena Vorrath, maria-elena.vorrath@awi.de

**Abstract**. In the last decades, changing climate conditions have had a severe impact on sea ice at the Western Antarctic Peninsula (WAP), an area rapidly transforming under global warming. To study the development of spring sea ice and environmental conditions in the pre-satellite era we investigated three short marine sediment cores for their biomarker inventory with particular focus on the sea ice proxy $IPSO_{25}$ and micropaleontological proxies. The core sites are located in the Bransfield Strait, in shelf to deep basin areas characterized by a complex oceanographic frontal system, coastal influence and sensitivity to large-scale atmospheric circulation patterns. We analyzed geochemical bulk parameters, biomarkers (highly branched isoprenoids, glycerol dialkyl glycerol tetraethers, sterols), as well as diatom abundances and diversity over the past 240 years, and compared them to observational data, sedimentary and ice core climate archives as well as results from numerical models. Based on biomarker results we identified four different environmental units characterized by (A) low sea ice cover and high ocean temperatures, (B) moderate sea ice cover with decreasing ocean temperatures, (C) high but variable sea ice

cover during intervals of lower ocean temperatures and (D) extended sea ice cover coincident with a rapid ocean warming. While $IPSO_{25}$ concentrations correspond quite well with satellite sea ice observations for the past 40 years, we note discrepancies between the biomarker-based sea ice estimates and the long-term model output for the past 240 years, ice core records and reconstructed atmospheric circulation patterns such as the El Niño Southern Oscillation (ENSO) and Southern Annular Mode (SAM). We propose that the sea ice biomarker proxies $IPSO_{25}$ and $PIPSO_{25}$ are not linearly related to sea ice cover and, additionally, each core site reflects specific, local environmental conditions. High $IPSO_{25}$ and $PIPSO_{25}$ values may not be directly interpreted as referring to high spring sea ice cover because variable sea ice conditions and enhanced nutrient supply may affect the production of both the sea-ice associated and phytoplankton-derived (open marine, pelagic) biomarker lipids. For future interpretations we recommend to carefully consider individual biomarker records to distinguish between cold, sea ice favoring and warm, sea ice diminishing environmental conditions.

**Key Words**: paleoclimate, Antarctic sea ice, highly branched isoprenoids, $IPSO_{25}$, diatoms, ENSO, SAM

## 1    Introduction

Observations of global mean surface temperatures show a warming of approximately $1.0\pm0.2°C$ (IPCC, 2018) above the 1850-1900 baseline as a result of progressive industrialization since the mid-19[th] century. An acceleration of this trend due to anthropogenic forcing has been projected (IPCC, 2019). The ocean, and especially the Southern Ocean, takes up the majority of the atmospheric heat, and warming has already been observed at all depths (IPCC, 2019). Antarctica´s hot spot of warming is the Western Antarctic Peninsula (WAP) (Jones et al., 2016) with an atmospheric temperature increase of $3.7\pm1.6°C$ in the 20[th] century (Vaughan et al., 2003) and a slight cooling from 2000 to 2010 (Turner et al., 2019). From the 1990s to 2000s, a warming of up to 1°C of subsurface water is also evident in different water masses around the WAP (Cook et al., 2016). On land, glaciers and ice shelves on both sides of the Antarctic Peninsula (AP) retreat rapidly since the 2000s (Cook et al., 2016; Rignot et al., 2019), pointing towards a potential collapse of the WAP ice shelves, while also the loss of sea ice cover is significant (Parkinson and Cavalieri, 2012). Shortened sea ice seasons (Parkinson, 2002) and a reduction of sea ice extent accelerating from 4 % up to 10 % per decade (Liu et al., 2004) have been observed in the region via satellite since 1979. A recent compilation shows that the steady increase in sea ice extent around the entire Antarctic continent since 1979 stopped in 2014 and was followed by fast decreases (Parkinson, 2019). However, the region along the WAP, the Bellingshausen Sea and Amundsen Sea show contrasting sea ice trends and high sea ice variability in 2014 and afterwards (Hobbs et al., 2016). The changes in sea ice cover are not only related to warm water intrusion and higher sea surface temperatures (SSTs) along the WAP (Martinson and McKee, 2012;

Meredith and King, 2005), but also to large-scale modes of atmospheric circulation such as the Southern Annular Mode (SAM) (e.g. Barbara et al., 2013) and the El Niño Southern Oscillation (ENSO) (e.g. Liu et al., 2004), or a combination of both (Etourneau et al., 2013; Stammerjohn et al., 2008b, 2008a).

Sea ice is an important factor that shapes and influences the Southern Ocean. Melting sea ice releases nutrients and leads to enhanced primary production and ocean stratification during spring and summer (Arrigo et al., 1997; Vernet et al., 2008). Interestingly, a higher number of sea ice days is associated with an increased photosynthetic efficiency and enhanced carbon fixation rates due to greater nutrient delivery stimulating primary production (Schofield et al., 2018) but also thinning of sea ice affects marine productivity positively (Hancke et al., 2018). Release of dense brine during sea ice formation leads to water mass transformations (Abernathey et al., 2016) that contribute to the thermohaline circulation by feeding of deep and intermediate waters (Nicholls et al., 2009) and at the same time inducing upwelling at sea ice edges (Alexander and Niebauer, 1981). Sea ice cover also reduces the ocean-atmosphere exchange of heat and gases as well as regional precipitation, enhances the albedo (Allison et al., 1982; Butterworth and Miller, 2016; Turner et al., 2017) and is a potential source of the radiative-relevant volatile dimethylsulphide produced by phytoplankton (Trevena and Jones, 2006) – a precursor of methanesulphonic acid (MSA) (Abram et al., 2010). Sea ice changes along the WAP may lead to the destabilization and/or collapse of local ice shelves due to warm water intrusions and basal melting (Cook et al., 2016; Etourneau et al., 2019; Hellmer et al., 2012) promoting an accelerated ice-sheet flow towards the ocean (Huss and Farinotti, 2014). Sea ice decline in this region may thus also indirectly contribute to global sea level rise.

Atmospheric circulation patterns such as ENSO and SAM have been suggested to control SST and the distribution of sea ice along the WAP (Ding et al., 2012; Stammerjohn et al., 2008b, 2008a). Etourneau et al. (2013) concluded from the occurrence of higher sea ice cover together with higher SSTs that a rising number of ENSO events would increase the seasonal amplitude of warmer summers and colder winters in the region. SAM is the leading climate mode in the Southern Hemisphere (Jones et al., 2016) and has significant impacts on temperatures at the northeast AP (Clem et al., 2016). Stammerjohn et al. (2008b) link ENSO and SAM related teleconnections to opposite sea ice trends in the Pacific and Atlantic sector of the Southern Ocean on decadal scales during the satellite era. The high-latitude responses and ice-atmosphere anomalies are strongest when a positive ENSO occurs "in-phase" with a negative SAM (+ENSO/-SAM) and the subtropical jet over the Pacific Ocean is strengthened whereas the polar frontal jet and the westerly winds are weaker (Stammerjohn et al., 2008b). In this state, a positive sea level pressure establishes a high-pressure cell in the Pacific Southern Ocean and warmer, moister conditions with less sea ice establish there. Meanwhile, the Weddell Sea and the WAP experience a cooling with an advance of sea ice. During the opposite state (-ENSO/+SAM) a stronger polar frontal jet establishes a low-pressure cell in the Bellingshausen

Sea. In this case, increased, south-ward migrated westerly winds transport heat towards the WAP and the Weddell Sea and sea ice cover is reduced under high atmospheric and sea surface temperatures (Marshall et al., 2006; Stammerjohn et al., 2008b; Yuan, 2004). Clem et al. (2016) propose that the combined effect of in-phase ENSO

and SAM is strongest in spring. A +SAM reduces the interaction of water masses between the Bransfield Strait and the Weddell Sea whereas higher Weddell Sea Water (WSW) input occurs during a -SAM due to the northward shift of the wind belt and ocean fronts and stronger coastal currents in the Weddell Sea (Dotto et al., 2016).

For modelling past and future Antarctic climate, ice sheet stability, the thermohaline circulation or the impacts of sea ice loss for ecosystems, data of past sea ice cover are crucial although barely available (Bracegirdle et al.,

2015, 2019). For the WAP, insights into climate and sea-ice dynamics during the past 200 years are available from ice cores (stable isotopes, marine aerosols and snow accumulation) but information from high resolution marine sediments and in particular sedimentary, geochemical or diatom-based sea ice proxies remain sparse (Thomas et al., 2019). Sinking marine particles carry environmental information from the sea surface to the ocean floor and, when buried in the sediments, the environmental history including sea ice can be deduced from these marine

climate archives. For sea ice reconstructions, the use of sea ice-associated diatom species and biogeochemical parameters are common (Crosta et al., 1998; Esper and Gersonde, 2014a; Gersonde and Zielinski, 2000). Since diatom frustules may be affected by the dissolution of biogenic opal in the photic zone (Ragueneau et al., 2000), on the ocean floor (Leventer, 1998) and in the sediments (Burckle and Cooke, 1983; Esper and Gersonde, 2014b), an increasing attention is paid to their molecular remains, i.e. specific biomarker lipids, as promising tools for past

sea ice reconstructions (Massé et al., 2011). A specific diunsaturated highly branched isoprenoid alkene (HBI diene, $C_{25:2}$) has been proposed as potential tool for past spring sea ice reconstructions in the Southern Ocean (Massé et al., 2011). It is produced by sea ice diatoms (Nichols et al., 1988) and its sea ice origin is evident from the high $\delta^{13}C$ isotopic signature of the molecule (Massé et al., 2011; Sinninghe Damsté et al., 2007; Vorrath et al., 2019). The sea ice diatom *Berkeleya adeliensis* which is observed to mainly occur during spring in Antarctic

landfast ice and platelet ice (Riaux-Gobin and Poulin, 2004) was identified as a producer of the HBI diene (Belt et al., 2016). The HBI diene is present in surface and downcore sediments around Antarctica and – in analogy to the Arctic IP$_{25}$ - can be used IPSO$_{25}$ (**I**ce **P**roxy for the **S**outhern **O**cean with **25** carbon atoms) likely showing the extent of near-coastal spring sea ice (Belt et al., 2016; Lamping et al., 2020; Massé et al., 2011; Riaux-Gobin and Poulin, 2004; Vorrath et al., 2019). To differentiate between an extended spring sea ice cover, the occurrence of a

stable sea ice margin and/or an open marine environment, IPSO$_{25}$ is combined with phytoplankton-derived biomarker lipids such as HBI trienes and/or sterols, which are considered as proxies of open water conditions (Belt and Müller, 2013; Volkman, 1986). Analogous to the PIP$_{25}$ index (P stands for open marine phytoplankton marker)

for semi-quantitative sea ice estimations in the Arctic (Müller et al., 2011), the recently proposed PIPSO$_{25}$ approach (Vorrath et al., 2019) allows for a differentiation between several sea ice conditions of a permanently open ocean, a sea ice marginal zone and a permanent sea ice cover.

Here, we provide the first IPSO$_{25}$-based high-resolution assessment of the spring sea ice development in the Bransfield Strait during the past 240 years and examine the response of sea ice to changes in atmospheric and oceanic oscillation patterns. To achieve this, we conducted a multiproxy study on three short sediment cores retrieved from different oceanic regimes to cover regional differences within the Bransfield Strait. In addition to IPSO$_{25}$, we analyzed HBI trienes, sterols and glycerol dialkyl glycerol tetraethers (GDGTs) for subsurface ocean temperature (SOT) reconstruction as well as diatom assemblages for estimating winter sea ice concentrations (WSI) and summer sea surface temperatures (SSST) by means of transfer functions. We furthermore consider sea ice and temperature data from an atmosphere-sea ice-ocean numerical model (AWI-ESM2), historical surface air temperatures from local meteorological stations, ice core records (stable isotopes $\delta^{18}$O and $\delta$D, MSA, annual net snow accumulation A$_n$), and paleo records of atmospheric circulation patterns such as ENSO and SAM.

## 2    Material and Methods

### 2.1    Study Area

The study area is the Bransfield Strait at the northern tip of the WAP (Fig. 1a and b). The region includes the shallow shelf of the WAP as well as the Bransfield Basin with depths exceeding 2000 m at its deepest parts. The Bransfield Basin is located between the South Shetland Islands (SSI) to the northwest and the AP to the southeast. The shallow ocean has been shaped by ice sheet dynamics during the last glaciation (Canals and Amblas, 2016b; Ingólfsson et al., 2003) and several troughs discharge sediment load from the AP and SSI into the basin (Canals et al., 2016; Canals and Amblas, 2016a). The oceanographic setting in this area is complex and not yet fully constrained (Moffat and Meredith, 2018; Sangrà et al., 2011) because water masses enter the basin from the west and east (Fig. 1b). From the east, relatively cold (< 0°C) and salty WSW flows at the surface alongshore the AP as a coastal current but also fills the Bransfield Basin completely below 150 m. It is also observed on the northern slope of the SSI at 200-600 m depth and around Elephant Island as a result of wind driven modulation (Meijers et al., 2016). The main water source from the west is the Bellingshausen Sea Water (BSW), transported by the Antarctic Circumpolar Current (ACC). This well-stratified, fresh and warm surface water flows along the slope of the SSI and forms the Peninsula Front with the WSW in the central Bransfield Strait, trending southwest-northeast parallel to the AP (Sangrà et al., 2011). Additionally, Circumpolar Deep Water (CDW) enters from the southwest as a subsurface current, forming the Bransfield Front to the BSW at 200m to 550m depth along the SSI slope

(Sangrà et al., 2017). Both BSW and CDW are observed to turn and flow back at the northern side of the SSI (Sangrà et al., 2011). The mixing and transformation of the three water masses in the Bransfield Strait is yet not well understood but a study of iceberg drifts from Collares et al. (2018) showed that water from the Weddell Sea join waters from the Bellingshausen Sea in the vicinity of Trinity Island (Fig. 1b). The input of WSW is suggested to be enhanced by a strong Weddell Gyre during a -SAM and diminished during a +SAM (Dotto et al., 2016). In the Bransfield Strait it has been suggested that eddies between the Peninsula Front and the Bransfield Front are a key mechanism for water exchange and/or upwelling (Sangrà et al., 2011; Zhou et al., 2002) and meltwater discharge from the adjacent glaciers has to be considered (Meredith et al., 2018). In the southwest, south of the Bransfield Strait, a narrow, fast flowing Antarctic Peninsula Coastal Current (APCC) is present, enriched in glacial freshwater and characterized by downwelling (Moffat and Meredith, 2018). The APCC surface water flow north- and southward of Anvers Island is significantly reduced during sea ice cover (Moffat and Meredith, 2018).

Primary productivity along the WAP is mainly controlled by eddies and fronts (Gonçalves-Araujo et al., 2015), due to upwelling (Sangrà et al., 2011), sea ice dynamics (Vernet et al., 2008) and iron distribution (Klunder et al., 2014). Diatom-associated high productivity regimes and high chlorophyll concentrations are found north of the Peninsula Front along the SSI under the influence of the BSW, while the area influenced by the WSW is characterized by lower production of nanoplankton (Gonçalves-Araujo et al., 2015). Heterogenous upwelling, iron fertilization and sea-ice retreat lead to high interannual variability in the production patterns and a strong onshore-offshore gradient is evident. In consequence high production is related to coastal areas, shallow mixed layers and higher stratification owing to sea ice melting (Sanchez et al., 2019; Vernet et al., 2008). High primary production is also reflected in high vertical export of sinking particles (e.g. Wefer et al., 1988; Kim et al., 2004) and in the biogeochemical distribution of surface sediments, dominated by high concentrations of TOC, pigments, sterols and diatoms but low calcium carbonate (Cárdenas et al., 2019). Organic matter is mainly of marine origin as supported by low values of C/N and the stable carbon isotope composition (Cárdenas et al., 2019) whereas the AP is an important source of terrestrial silts and clays (Wu et al., 2019).

### 2.2    Sampling and age model

The cores were collected in 2016 during the RV *Polarstern* cruise PS97 (ANT-XXXI/3) using a multicorer at stations PS97/056-1 (63°45.42'S, 60°26.51'W, 633 m water depth) east of Trinity Island, PS97/068-2 (63°10.05'S, 59°18.12'W, 794 m water depth) in the Orleans Trough, and PS97/072-2 (62°00.39'S, 56°03.88'W, 1992 m water depth) in the East Bransfield Basin (Fig. 1b). Smear slides were examined and microscopic description of the surface sediments was done onboard (Lamy, 2016). Immediately after recovery, sediment cores were sectioned into 1 cm slices and subsampled onboard. Samples designated for biomarker analyses were stored in glass vials at

-20° C and samples for micropaleontological analyses were stored in plastic bags at +4° C. A second suite of samples from a trigger core from station PS97/072-1 was used for diatom analyses (diatom samples from core PS97/072-2 were not available).

Geochronology for the sediment cores from sites PS97/056-1 and PS97/072-2 was established using $^{210}Pb_{xs}$ activities quantified by alpha spectrometry of its daughter $^{210}Po$ in secular equilibrium with $^{210}Pb$ and using $^{209}Po$ as a yield tracer (Flynn, 1968). The activities were corrected to the time of plating considering the $^{210}Po$ decay (half life: 138 days). $^{210}Pb_{xs}$ (unsupported) activities were determined as the difference between $^{210}Pb$ and $^{226}Ra$ activities measured by gamma spectrometry in some intervals of the sediment core. Alpha and gamma counting were performed at the Laboratoire Géosciences of the Université de Montpellier (France). The ages were based on $^{210}Pb_{xs}$ inventories according the Constant Rate of Supply Model (CRS, Appleby and Oldfield, 1978). Standard deviations (SD) were estimated propagating the counting uncertainties (Bevington et al., 1993). Since the dating of cores PS97/056-1 and PS97/072-2 was done on selected samples the age model was established using the software R (R Core Team, 2012) and the package clam (Blaauw, 2010, version 2.3.2, calibration curve Marine13.14C). Trigger core PS97/072-1 was correlated to the age model of core PS97/072-2 based on TOC data (see supplement S1).

$^{210}Pb_{xs}$ for core PS97/068-2 was measured at the Alfred Wegener Institute (AWI, Germany) on dried and ground bulk sediment samples in sealed gas-tight petri dishes, using a HPGe gamma spectrometer with planar geometry. $^{210}Pb$ was measured at 46 keV and $^{226}Ra$ was measured for the excess correction in each depth interval via its indirect decay products at 295, 352 and 609 keV. Analytical errors were calculated considering error propagation. For core PS97/068-2 the calculation of CRS ages and the Monte-Carlo approximation of age uncertainties was based on Sanchez-Cabeza et al. (2014), modified to accommodate the variable sample sizes and fractions for different depths. Due to residual inventory of $^{210}Pb_{xs}$ below the available samples in cores PS97/056-1 and PS97/072-2, the CRS model had increasing uncertainties below ~130 years (supplement S2). We therefore extrapolated ages before 1880 based on the average respective sedimentation rates for the oldest 3 cm (Fig. 2).

**2.3 Organic geochemical analyses**

Organic geochemical analyses were done on freeze-dried and homogenized sediments. Bulk content of carbon (C) and nitrogen (N) were determined on 30 mg sediment sample with a CNS analyzer (Elementar Vario EL III, standard error < 2%), whereas the analysis of TOC content was done on 0.1 g acidified (500 µl hydrochloric acid) sediment sample using a carbon-sulphur determinator (CS-2000, ELTRA, standard error < 0.6 %). The C/N ratio was calculated as TOC/total nitrogen.

The extraction procedure of HBIs follows the analytical protocol of the international community conducting HBI-based sea ice reconstructions (Belt et al., 2013, 2014; Stein et al., 2012). For the quantification of biomarkers the internal standards 7-hexylnonadecane (7-HND; HBI standard), 5α-androstan-3β-ol (sterol standard) and $C_{46}$ (GDGT standard) were added to the sediments. Sediment samples of 5 g were processed ultrasonically three times using 6 ml of $CH_2Cl_2$:MeOH (v/v 2:1, 15 min) followed by centrifugation (2500 rpm, 1 min) and decantation of the total organic solvent extract. The different biomarkers were separated via open column chromatography with silica gel used as a stationary phase. First, the apolar fraction containing HBIs was separated with 5 ml hexane, while the second polar fraction containing GDGTs and sterols was eluted with 5 ml $CH_2Cl_2$:MeOH (v/v 1:1). The polar fraction (GDGT and sterols) was dried using nitrogen, re-dissolved in 120 µl hexane:isopropanol (v/v 99:1) and filtered through a polytetrafluoroethylene filter (0.45 µm in diameter). After measuring GDGTs, the polar fraction of the sample was silylated (200 µl BSTFA; 60° C; 2 hours) and used for sterol analysis.

The HBIs and sterols were analyzed by GC-MS with an Agilent 7890B gas chromatograph equipped with a 30 m DB 1 MS column (0.25 mm diameter, 0.250 µm film thickness) and coupled to an Agilent 5977B mass spectrometer (MSD, 70 eV constant ionization potential, ion source temperature 230° C). Apolar and polar lipid fractions were analyzed using different temperature programs. For HBIs, the temperature was held at 60° C for 3 min, gradually increased to 325° C over the course of 23 min, and was sustained at this level for 16min. Sterol analysis started at a temperature of 60° C for 2 min, the temperature then gradually increased to 150° C over the course of 6 min, and continued to increase to 325° C over a course 57 min. HBIs were identified via comparison of mass spectra of the measured compounds and published mass spectra (Belt et al., 2000). Quantification of HBIs was based on manual peak integration. Instrumental response factors of molecular ions of HBI diene (*m/z* 348) und trienes (*m/z* 346) were determined by means of calibration measurements using a sample with known concentrations of HBIs. Identification of sterols was based on comparison of their retention times and mass spectra with those of reference compounds analyzed on the same instrument. The mean relative error of duplicates was < 5% for HBIs and < 1% for sterols (desmosterol had exceptional high relative errors of up to 14%), the detection limit was determined at 0.5 ng/g sediment. Co-elution of other compounds hampered identification and quantification of sterols in several samples (PS97/056-1; 0-13cm and PS97/072-2; 0-16cm). Concentrations of HBIs and sterols were normalized to TOC contents (µg/g TOC).

GDGTs were analyzed using high performance liquid chromatography (HPLC, Agilent 1200 series HPLC system) coupled to a single quadrupole mass spectrometer (MS, Agilent 6120 MSD) via an atmospheric pressure chemical ionization (APCI) interface. Individual GDGTs were separated at 30° C on a Prevail Cyano column (150 mm x 2.1 mm, 3 µm). Each sample was injected (20 µl) and passed a 5 min isocratic elution with mobile phase A

(hexane/2-propanol/chloroform; 98:1:1) at a flow rate of 0.2 ml/min. The mobile phase B (hexane/2-propanol/chloroform; 89:10:1) increased linearly to 10% within 20 min and after this to 100% within 10 min. After 7 min the column was cleaned with a backflush (5 min, flow 0.6 ml/min) and re-equilibrated with solvent A (10 min, flow 0.2 ml/min). The APCI had the following conditions: nebulizer pressure 50 psi, vaporizer temperature 350°C, $N_2$ drying gas temperature 350°C, flow 5 l/min, capillary voltage 4 kV, and corona current 5 μA. GDGT detection was done by selective ion monitoring (SIM) of (M+H)$^+$ ions (dwell time 76ms). The molecular ions $m/z$ of GDGTs-I ($m/z$ 1300), GDGTs-II ($m/z$ 1298), GDGTs-III ($m/z$ 1296), and Crenarchaeol ($m/z$ 1292) as well as of the branched GDGTs-Ia ($m/z$ 1022), GDGTs-IIa ($m/z$ 1036), GDGTs-IIIa ($m/z$ 1050) and hydroxylated GDGTs OH-GDGT-0 ($m/z$ 1318), OH-GDGT-1 ($m/z$ 1316), and OH-GDGT-2 ($m/z$ 1314) were quantified in relation to the internal standard $C_{46}$ ($m/z$ 744). The hydroxylated GDGTs were quantified in the scans of their related GDGTs (see Fietz et al., 2013). The standard deviation was 0.01 units of $TEX^L_{86}$.

We follow the recommendation of Fietz et al. (2020) and apply both hydroxylated and non-hydroxylated GDGT temperature estimations because their significance for different ocean regions is still a subject of many discussions (Fietz et al., 2016; Huguet et al., 2013; Liu et al., 2012; Lü et al., 2015; Schouten et al., 2013) especially for low temperature paleo events at continental margins (Wang et al., 2015). After Kalanetra et al. (2009) GDGT-derived temperatures represent near-surface waters which is underlined by studies from Kim et al. (2012) and Park et al. (2019) and therefore we consider our results to reflect subsurface ocean temperatures (SOT). For calculation of $TEX^L_{86}$ (Kim et al., 2010) only GDGTs with the $m/z$ 1296 (GDGT-3), $m/z$ 1298 (GDGT-2), $m/z$ 1300 (GDGT-1) were considered in Eq. (1):

$$TEX^L_{86} = \log \left( \frac{[GDGT-2]}{[GDGT-1]+[GDGT-2]+[GDGT-3]} \right) \tag{1}$$

and calibrated it with Eq. (2) $SOT^{TEX} = 67.5 \times TEX^L_{86} + 46.9$ (Kim et al., 2010). $\tag{2}$

The calculation based on OH-GDGT was done after Lü et al. (2015) in Eq. (3)

$$RI - OH' = \frac{[OH-GDGT-1]+2\times[OH-GDGT-2]}{[OH-GDGT-0]+[OH-GDGT-1]+[OH-GDGT-2]} \tag{3}$$

and calibrated with Eq. (4) $SOT^{OH} = (RI\text{-}OH' - 0.1) / 0.0382$. $\tag{4}$

We assume that both $SOT^{TEX}$ and $SOT^{OH}$ reflect annual mean temperatures because their calibrations are based on annual mean SST (Kim et al., 2010; Lü et al., 2015).

To determine the influence of terrestrial organic matter the BIT index was calculated following Hopmanns et al. (2004) as Eq. (5)

$$BIT = \frac{[GDGT-Ia]+[GDGT-IIa]+[GDGT-IIIa]}{[Crenarchaeol]+[GDGT-Ia]+[GDGT-IIa]+[GDGT-IIIa]}. \tag{5}$$

The phytoplankton-IPSO$_{25}$ index (PIPSO$_{25}$) was calculated following Eq. (6) from Vorrath et al. (2019) with

$$275 \quad PIPSO_{25} = \frac{IPSO_{25}}{IPSO_{25} + (c \times phytoplankton\ marker)} \qquad (6)$$

using sterols and HBI trienes as phytoplankton marker (Vorrath et al., 2019). The balance factor c (c = mean $IPSO_{25}$ / mean phytoplankton biomarker) is used to account for concentration offsets between $IPSO_{25}$ and phytoplankton biomarkers (Belt and Müller, 2013; Müller et al., 2011; Smik et al., 2016b; Vorrath et al., 2019). Since the concentrations of HBI trienes are within the same range as the sea ice proxy we set the c-factor to 1

(Smik et al., 2016b) and c-factors for sterols were calculated individually for every core site. To distinguish the different indices based on their phytoplankton marker we use the terms $P_Z IPSO_{25}$ for an index based on Z-trienes, $P_E IPSO_{25}$ based on E-trienes, $P_B IPSO_{25}$ based on brassicasterol, and $P_D IPSO_{25}$ based on dinosterol.

**2.4    Diatom analysis and transfer functions**

Diatom analyses were done on 94 samples in total. Every second centimeter of core PS97/056-1 and every

centimeter of core PS97/068-2 and trigger core PS97/072-1 was analyzed. About 300 mg of freeze-dried sediments were treated after the method described by Cárdenas et al. (2019) and slides for microscopy analysis were prepared after Gersonde and Zielinksi (2000). Two permanent slides per sample were prepared and observed with a Carl Zeiss Axio Lab.1 microscope with phase contrast at $1000\times$ magnification at the Instituto Antártico Chileno in Punta Arenas.  Diatoms were identified and counted on transects on microslides until reaching at least 400 valves

on each slide, following counting procedures of Schrader and Gersonde (1978). Diatom identification was done to species or species group level following the taxonomy described by Armand and Zielinski (2001), Taylor et al. (2001), Crosta et al. (2004), Buffen et al. (2007), Cefarelli et al. (2010), Esper et al. (2010), Allen (2014), and Campagne et al. (2016). The Hyalochaete of the genus *Chaetoceros* were identified as vegetative cells and/or resting spores.

We applied the marine diatom transfer function TF MAT-D274/28/4an to estimate winter sea ice (WSI) concentrations. It comprises 274 reference samples, including two samples from the AP, with 28 diatom taxa and/or taxonomic groups and an average of 4 analogues from surface sediments that cover the complete circumpolar-Antarctic in the Atlantic, Pacific, and western Indian sectors of the Southern Ocean (Esper and Gersonde, 2014a). Winter sea ice (WSI) estimates reflect September sea-ice concentrations averaged over the

period from 1981-2010 (National Oceanic and Atmospheric Administrations, NOAA; Reynolds et al., 2002, 2007) in a 1 by 1 grid. We follow the approach of Zwally et al. (2002) and define a sea ice concentration of 15% as the threshold for presence or absence of sea ice and 40% as the representative average of sea-ice edge (Gersonde et al., 2005; Gloersen et al., 1993). For summer sea surface temperature (SSST), we used the transfer function TF IKM336/29/3q from 336 reference samples (covering all sectors of the Pacific, Atlantic and Indian Southern

Ocean, 4 samples from the AP) with 29 diatom taxa and three factors (Esper and Gersonde, 2014b). For

calculations of MAT and IKM the software R (R Core Team, 2012) was used with the packages Vegan (Oksanen et al., 2012) and Analogue (Simpson and Oksanen, 2012).

## 2.5 Modelled data

We used data from numerical modelling to compare and evaluate our biogeochemical proxies. The AWI-ESM2 is a state-of-the-art coupled climate model developed by Sidorenko et al. (2019). The model consists of the atmospheric model ECHAM6 (Stevens et al., 2013) and the finite element sea ice-ocean model (FESOM2) (Danilov et al., 2017). It also includes a Land-Surface Model (JSBACH) with static ice sheets and dynamical vegetation (Raddatz et al., 2007).

The atmosphere grid in the high-resolution experiment is T63 (about 1.9 degree or 210 km) with 47 vertical levels. A multi-resolution approach is employed in the ocean module. In detail, the high-resolution experiment applies up to 20 km horizontal resolution over the Arctic and Antarctic region and 150 km for the far field ocean (supplement S3). Moreover, the tropical belt has a refined resolution of 30-50 km in this configuration. We used a fixed ice sheet and the topography was taken from ICE6G (Peltier et al., 2015) so isostatic adjustments are neglected. There are 46 uneven vertical depths in the ocean component. The model has been validated under modern climate condition (Sidorenko et al., 2019). Previous versions of the model have been applied for the Holocene (Shi et al., 2020; Shi and Lohmann, 2016).

We run the climate model from the Mid-Holocene as a starting point (*midHolocene* simulation), and performed a transient simulation from the Mid-Holocene to the pre-industrial (*past6k* simulation) along the recipe as described in Otto-Bliesner et al. (2017). The transient orbital parameters are calculated according to Berger (1978). Moreover, as the change of topography from mid-Holocene to present is minor, we use constant topography under pre-industrial conditions for the entire transient period. In our modeling strategy, we follow Lorenz and Lohmann (2004) and use the climate condition from the pre-industrial state as spinup and initial state for the transient simulation covering the period 1850-2017 CE. Greenhouse gases concentrations are taken from the ice core records (Köhler et al., 2017) and from Meinshausen et al. (2011).

## 2.6 Additional data sets

Regional monthly satellite sea ice concentrations were derived from Nimbus-7 SMMR and DMSP SSM/I-SSMIS passive microwave data from the National Snow and Ice Data Center (NSIDC, grid cell size 25x25 km, Cavalieri et al., 1996) and mean winter (JJA) and spring (SON) sea ice concentrations were used in this study.

For the large-scale atmospheric modes we used the paleo ENSO index from Li et al. (2013), which covers the past 200 years and the modelled SAM data from Abram et al. (2014).

We used ice core stable isotope data representing relative air temperature at James Ross Island ($\delta$D, Abram et al., 2013) and at Bruce Plateau ($\delta^{18}$O, Goodwin et al., 2016). We compared the marine sea ice proxies (biomarkers, diatoms) with MSA data from the West Antarctic Dyer Plateau ice core (Abram et al., 2010) and annual net snow accumulation from Bruce Plateau ($A_n$, Goodwin et al., 2016).

## 3    Results

### 3.1    Age model and core description

The $^{210}$Pb signals indicated continuously increasing ages with depth in all sediment cores (Fig. 2). All sediment cores roughly cover the last 240 years (including the extrapolated time) with resolution between 2 and 12 years per centimeter and sedimentation rates from 0.1 to 0.5 cm/a. Core PS97/056-1 located east of Trinity Island is characterized by silt-bearing diatomaceous clay (Lamy, 2016) and covers the timespan from 1830 to 2006 CE with sedimentation rates increasing from 0.1 to 0.4 cm/a over time. Core PS97/068-2 from Orleans Trough consists mainly of diatom-bearing silty clay (Lamy, 2016) and spans from 1780 to 2007 CE with sedimentation rates from 0.1 to 0.5 cm/a. Sediment core PS97/072-2 from the East Bransfield Basin is record from the greatest depth and characterized as silt-bearing diatomaceous clay (Lamy, 2016) with increasing sedimentation rates (from 0.1 to 0.4 cm/a) covering the time from 1823 to 2000 CE. At all three cores sites, sedimentation rates increase towards the present. The TOC contents of all cores ranged between 0.7 and 1.1 wt% with higher contents in younger sediments. Low C/N ratios (< 8.6) and BIT values (< 0.02) point to a marine origin of the organic matter.

### 3.2    Biomarker lipids

A summary of biomarker results is visualized in Figure 3 (results of HBI E-trienes, sterols and their related sea ice indices can be found in Figure S4 in the supplements). IPSO$_{25}$ is abundant at all core sites with values ranging from 0.2 µg g$^{-1}$ TOC up to 6.4 µg g$^{-1}$ TOC. All three cores display similar patterns with low IPSO$_{25}$ concentrations before 1850 CE followed by moderate concentrations until 1970 CE and maxima in the 2000s (Fig. 3). Concentrations of HBI trienes are much lower than IPSO$_{25}$ concentrations with values below 1.4 µg g$^{-1}$ TOC for Z-trienes (Fig. 3) and below 0.7 µg g$^{-1}$ TOC for E-trienes (supplement S4). The exception is core PS97/072-2 from of the East Bransfield Basin where both HBI trienes reach up to 3.7 µg g$^{-1}$ TOC and 1.6 µg g$^{-1}$ TOC, respectively, in the second half of the 19$^{th}$ century. The concentrations of brassicasterol (10.2–241.3 µg g$^{-1}$ TOC) and dinosterol (5.0–145.2 µg g$^{-1}$ TOC) are two to three magnitudes higher than those of the HBIs; markedly lower concentrations characterize the Orleans Trough (PS97/068-2) (supplement S4). The PIPSO$_{25}$ indices calculated with Z- and E-trienes run parallel to PIPSO$_{25}$ based on brassicasterol and dinosterol and show increasing trends with time. In general, the HBI triene-based PIPSO$_{25}$ indices have higher values (P$_Z$IPSO$_{25}$ from 0.32 to 0.91; P$_E$IPSO$_{25}$ from

0.25 to 0.95) than the PIPSO$_{25}$ indices based on sterols (P$_B$IPSO$_{25}$ from 0.15 to 0.70; P$_D$IPSO$_{25}$ from 0.11 to 0.75). The PIPSO$_{25}$ indices suggest an increasing spring sea ice cover over time (Fig. 3, supplement S4). This is most prominent at the East Bransfield Basin (PS97/072-2) where lowest sea ice cover is indicated around 1870 CE and increase towards the present is indicated. Indications of short-term low spring sea ice cover are found for the 1960s

and 1970s at the near-coastal core sites (PS97/056-1 and PS97/068-2) but do not change the overall trend. A clear stratigraphy is hard to distinguish so we focus on units that reflect similar sea ice conditions taken from our sea ice proxies (Fig. 3). We clearly stress out that our age models were extrapolated before 1880 and hence, the age uncertainty increase in this period.

Temperatures based on GDGTs show a wide range of values. At Trinity Island (PS97/056-1) and the East

Bransfield Basin (PS97/072-2), SOT$^{TEX}$ range from -3.89°C to 2.3°C (Fig. 3) whereas temperatures are always above zero from 0.7° C to 3.6° C at the Orleans Trough (PS97/068-2). Distinct cold events occur in the 1860s in the East Bransfield Basin (PS97/072-2) and as a longer cool period from 1940 to 1970 CE at the coastal core sites but general trends are hard to distinguish. In contrast, SOT$^{OH}$ displays a decreasing temperature trend at all core sites with a narrow range of -2.56 °C to -1.0° C reversed by a rapid warming since the 1990s (Fig. 3). As some

temperatures lie below the freezing point of sea water we assume that neither SOT$^{TEX}$ nor SOT$^{OH}$ may reflect exact temperatures but temperature trends at each core site.

### 3.3 Diatom assemblages

Winter sea ice estimations derived from diatom assemblages point to a high variability (74% to 92% WSI at PS97/056-1, 64% to 92% at PS97/068-2, 68% to 90% at PS97/072-1) with minimum sea ice concentrations around

385 1840 and 1880 CE and a slight increment toward 1990s (Fig. 3). This variability coincides with the abundances of the sea ice diatom species *Fragilariopsis curta* that show higher values in cores PS97/056-1 (abundance range 0.1-1.8%) and 068-2 (abundance range 2.0-10.7%) in the second half of the 20th century (Fig. 3) while no trends are present in PS97/072-1 (abundance range 1.9-7.6%). In addition, WSI records reveal similar features compared to IPSO$_{25}$ and PIPSO$_{25}$, which points to an expected coherence of winter and spring sea ice estimates based on

different proxies. The SSST from diatom assemblages have a small amplitude in all cores (-0.9 to 0.5°C at PS97/056-1, -1.1 to 0.2°C at 068-2 and -0.8 to 0.1°C at 072-1) and show a similar pattern to SOT$^{TEX}$ at the sites PS97/068-2 and 072-1 (Fig. 3).

### 3.4 Modelled data

We use model data as derived from the AWI-ESM2 which include spring sea ice concentration (mSSIC), spring

sea ice thickness (mSSIT), subsurface ocean temperature (mSOT, mean temperature from 30-100 m below sea surface), and surface air temperature (mSAT). Based on 10-year means, we detect negative trends for the last 200

years in both mSSIC (decrease by 30% to 50%) and mSSIT (decrease from 0.5 m down to 0.1 m). At the same time, positive trends for mSOT and mSAT at all core sites show temperatures rising by 0.3°C to 0.6°C. Further, a time series of the latitudinal shift of the sea ice edge at the WAP (between 50°W and 70°W) which shows a southward shift of 1.5° from 61.9°S to 63.4°S in the 20th century.

## 4 Discussion

### 4.1 Spatial and temporal distribution of paleoenvironmental biomarkers

The core site at Trinity Island (PS97/056-1) is dominated by the APCC and receives freshwater input from the Peninsula with influence of BSW from the ACC (Moffat and Meredith, 2018). We suggest that sea ice proxies originate from free floating or land fast sea ice and are impacted by meltwater discharge in this region since the core site is only 8 km away from Trinity Island. Coastal upwelling of macro- and micronutrients, especially iron, and a stratified water column fuel open marine primary production (Sanchez et al., 2019; Vernet et al., 2008) and may explain highest concentrations of sterols at this core site. $IPSO_{25}$, HBI Z-triene, $P_ZIPSO_{25}$, WSI and *F. curta* records exhibit similar trends and fluctuations over time (Fig. 3) but a direct relation between reconstructed sea ice conditions and temperature (SSST, $SOT^{TEX}$ and $SOT^{OH}$) is not evident. However, slightly higher temperatures deduced from $SOT^{OH}$ and diatom-SSST seem to coincide with lower $IPSO_{25}$ concentrations, lower $PIPSO_{25}$ values and reduced WSI and *F.curta* content in the 19th century, while variable but higher temperatures in the 20th century are accompanied by higher $IPSO_{25}$ and WSI concentrations at site PS97/056-1 (Fig. 3). The remarkably low $SOT^{TEX}$ in the year 2006 CE might be a result of cold meltwater injections due to enhanced glacier melting (e.g. Pastra Glacier on Trinity Island). At the same time high SSST and $SOT^{OH}$ as well as a low WSI point towards a significantly warm period around the year 2006 which is underlined by meteorological station data (Turner et al., 2019). A weak cooling trend is present in SSST and $SOT^{OH}$ from 1920 CE to the 1990s, followed by a warming towards the present.

Given the position of the sediment core PS97/068-2 in the Orleans Trough we suggest that the core site is affected by the Peninsula Front where water masses from both salty and cold WSW and fresh and warm BSW meet. The water column here is characterized by enhanced mixing within a narrow eddy zone and deepening of the mixed layer (Sangrà et al., 2011). High concentrations of biomarkers indicate a strengthening of primary productivity associated with BSW (Gonçalves-Araujo et al., 2015) in a less stratified and mixed water column (Vernet et al., 2008). The patterns of $IPSO_{25}$, HBI Z-triene as well as $P_ZIPSO_{25}$, WSI and *F.curta* have a good visual correspondence. They indicate higher phytoplankton productivity and higher sea ice cover towards the present time. The $SOT^{TEX}$ is remarkably high (above 0° C) throughout the studied period contrasting modern ocean

temperatures that are below -0.5° in the upper 400 m along the WAP (Cook et al., 2016). Assuming that enhanced mixing of water masses with the atmosphere leads to warmer ocean temperatures, this must be evident in our other temperatures records too. However, since none of the other temperature proxies show such a response we must treat this record with caution. We note that OH-based records seem to better reflect temperature estimations in polar regions (Fietz et al., 2020) and suppose that maybe water mixing might have a disturbing effect of TEX-based temperature reconstructions. Compared to SOT$^{TEX}$, SOT$^{OH}$ temperatures are closer to modern ocean temperatures in this area (-0.5°C, Cook et al., 2016) within a narrow range. Since this core site is located at the Peninsula Front (i.e. in the middle of BSW and WSW), no clear dominance of one or the other water mass is evident. Ocean temperatures below 0°C indicate the influence of WSW while high biomarker concentrations refer to higher primary production as it is suggested for the BWS (Gonçalves-Araujo et al., 2015).

The core site in the East Bransfield Basin (PS97/072-2) is further away from the coast (145 km) compared to the other two core sites. Marine productivity is expected to be lower due to the presence of WSW (Gonçalves-Araujo et al., 2015) but relatively high concentrations of IPSO$_{25}$ and HBI Z-triene may be related to fertilization through iron input from shelf waters (Frants et al., 2013). A remarkable maximum in HBI Z-triene concentrations in the late 19$^{th}$ century suggests drastic changes in the local oceanographic settings and productivity patterns. As indicated by SOT$^{TEX}$, this period is marked by a rapid shift from cold to warm subsurface ocean temperatures, pointing to a possible dominance of warmer BSW. A corresponding retreat of sea ice cover and likely ice-free summers, as reflected by P$_Z$IPSO$_{25}$ and WSI values, could have promoted the productivity of open marine or coastal phytoplankton communities, e.g. *Rhizosolenia* and *Pleurosigma*, synthesizing the HBI Z-triene (Belt et al., 2000, 2017). Sea ice proxies show less pronounced increasing trends at this core site compared to the near-coastal records. At all three core sites the sedimentation rates increase towards present. Together with higher TOC contents we suggest this to be linked to a higher export of organic matter although primary production tended to decrease in the past 30 years (Montes-Hugo et al., 2009).

We note that for the interpretation of biomarker-based sea ice reconstructions the potential degradation of biomarkers affecting their downcore concentration profile needs to be taken into consideration. We observe that the upper part of the sediment cores contains higher concentrations of IPSO$_{25}$, HBI trienes and sterols compared to the underlying older sediments. A similar pattern in IPSO$_{25}$ and HBI triene concentrations is also reported by Barbara et al. (2013). Their biomarker concentrations from the southern WAP equal the concentrations in the Bransfield Strait but high values near the sediment surface, as in our data, are not present. Auto- and photooxidative degradation of IPSO$_{25}$ and HBI trienes was already studied in laboratory experiments (Rontani et al., 2014, 2011) and autoxidative and bacterial degradation was also found in the oxic layers of surface sediments (Rontani et al.,

2019). According to these results, a faster degradation of HBI trienes (because of their higher number of double bonds) in the upper centimeters of the herein studied sediment cores would lead to higher PIPSO$_{25}$ values with progressive degradation. Sterols might also undergo degradation (Rontani et al., 2012) but studies from Antarctic sediments are still missing. However, as we observe remarkably high HBI triene concentrations in core PS97/072-2 in the late 20$^{th}$ century and lower concentrations towards the present (Fig. 3, supplement S4), we suggest that degradation may not have a major impact on biomarker concentrations presented in this study.

### 4.2 Comparison of proxy-derived and modelled sea ice estimates with satellite sea ice observations

We compare IPSO$_{25}$ concentrations, P$_Z$IPSO$_{25}$ values, diatom-based WSI estimates and *F.curta* content with satellite data and with mSSIC to evaluate their accuracy in reflecting spring and winter sea ice cover variability at the core sites over the past 40 years (Fig. 4). Satellite-derived spring sea ice concentrations (satSSIC) show general similarities to fluctuations observed in the IPSO$_{25}$ record indicating lower sea ice cover in the 1980s, a peak in the mid-1990s, a drop in sea ice cover in the early 2000s and then again a rise in sea ice concentrations (Fig. 4).

At the near-coastal core sites (PS97/056-1 and 068-2), these dynamics are well reflected in IPSO$_{25}$ and PIPSO$_{25}$, in particular for site PS97/056-1, where a good correspondence is observed between biomarker and satellite data (Fig. 4). However, we cannot exclude aliasing effects due to a lower temporal resolution of the sediment cores (Pisias and Mix, 1988). Maximum sea ice concentrations observed in 1996 CE, are well reflected by elevated IPSO$_{25}$ concentrations but the drop afterwards is not mirrored by biomarker data. Diatom-based WSI compared to satellite-derived winter sea ice concentrations (satWSIC) show that these two data sets are in moderate agreement at the near-coastal core sites (PS97/056-1 and PS97/068-2) and winter sea ice coverage seems to be less dynamic at the Peninsula Frontal mixing zone (PS97/068-2). It was already observed that WSI indicated much higher sea ice cover than satellite data (Vorrath et al., 2019) as the transfer function for WSI is build on a different satellite reference data set and rather represents sea ice probability than sea ice cover (Esper and Gersonde, 2014a). We also note that the modelled spring sea ice cover is mostly opposite to satellite data, in particular during the 1990s and 2000s. Such bias is found to be common in recent sea ice modelling and can be traced back to the model's sensitivity to warming (Rosenblum and Eisenman, 2017) and the role of cloud feedbacks (Bodas-Salcedo et al., 2014). While modelled and satellite derived data have similar ocean grid sizes (model: 30x30 km, satellite: 25x25 km) we suggest that global models such as AWI-ESM2 cannot resolve the AP sub-aerial and marine topography and have difficulties in capturing local to regional near coastal sea-ice dynamics in the study region. Another reason may relate to internal variability and missing feedbacks in the model which makes a direct comparison of short time series difficult. Changes in the forcing are restricted to the insolation and greenhouse gases and can affect the simulated climate by bringing in natural noises. For the 240 years of modelled period, especially for

small changes in the forcing, the internal variabilities can dominate the climate change bringing difficulties to

490 model-data comparisons. Feedbacks of aerosols, ozone, ice sheet dynamics, dust, solar and volcano activity are

missing because these elements were considered static in the model. Further, the modelled Antarctic sea ice is

generally thicker and the coverage is higher due to a reduced warming of the Southern Ocean within the model

setup (Sidorenko et al., 2019). However, both modelled and satellite data for all core sites show a decreasing trend

in spring sea ice cover (about 10%) and a slightly rising trend in winter sea ice cover over the 40 year period (about

495 7%), while sea-ice proxies suggest an increasing trend of spring sea ice. For winter sea ice, core sites PS97/056-1

and PS97/072-2 display a decreasing trend, whereas PS97/068-1 clearly points to an increase in winter sea ice.

For the offshore core site at the East Bransfield Basin (PS97/072-2), $IPSO_{25}$ and $PIPSO_{25}$ correspond better to

satSSIC than to mSSIC sea ice data (Fig. 4). Between 1985 and 1995 CE, both $PIPSO_{25}$ indices suggest a similar

increase in spring sea ice as the satellite observations. Sea ice estimates from WSI and *F.curta* seem to better

reflect satSSIC than satWSIC. Also, WSI estimates are remarkably lower than at the other core sites, although

satellite winter sea ice cover is the highest of all.

Based on the overall accordance with satellite data, we conclude that the biomarker and diatom-based sea ice

estimations are related to regional dynamics of sea ice cover, as far as we can assess it from the low resolution of

the sediment cores. Regarding the oceanographic setting, we consider that also drift ice originating in the Weddell

Sea may have affected the deposition of $IPSO_{25}$. Since HBI Z-trienes and sterol concentration profiles are similar

to $IPSO_{25}$ concentrations (Fig. 3, supplement S4) we suggest that sea ice dynamics also promote growth of open

marine phytoplankton species due to nutrient release or nutrient upwelling (Sanchez et al., 2019; Vernet et al.,

2008). Input of allochthonous material from the shelf via near bottom nepheloid layers is also possible (Palanques

et al., 2002), which might impact the fidelity of the proxy records. As the record of satellite observations is short,

it is not clear whether decadal or centennial sea ice trends can be directly derived from our biomarker records.

Also, the resolution of our sediment records is quite low (3 to 5 years) compared to satellite observations. Hence,

we consider modelled and ice core data for further insights over the full sediment records.

**4.3    Comparison of sea ice proxy records with modelled and ice core data covering the pre-satellite era**

By comparing $IPSO_{25}$, $P_ZIPSO_{25}$-based sea ice estimates, WSI and *F.curta* with modelled spring sea ice data, we

note opposite long-term sea ice trends reflected in the proxy records and the modelled data for the past 240 years

(Fig. 5). Modelled spring sea ice concentration and thickness show clear decreasing trends at all sites with a loss

of sea ice cover between 15% and 20%. Modelled sea ice cover fluctuates strongly in the East Bransfield Basin

(PS97/072-2) whereas the coastal core sites run almost parallel. Although the modelled spring sea ice does not

agree with satellite data on local to regional scale (Fig. 4) it does reflect the satellite observations regarding the

large-scale general trend of sea ice decline and warming in the Bellingshausen Sea and at the WAP (Parkinson and Cavalieri, 2012; Vaughan et al., 2003) because the model is based on rising greenhouse gas concentrations in the 20[th] century.

The increasing concentrations of IPSO$_{25}$ as well as the rise of parallel running P$_Z$IPSO$_{25}$ values, diatom-derived WSI concentrations and rising *F.curta* content recorded in all three sediment cores suggest a long-term sea ice advance. On the other hand, the increase in the concentrations of the HBI Z-triene and sterols would indicate more open marine and/or stable ice edge conditions promoting phytoplankton productivity. We suggest that a thinning of the ice and a hence higher light penetration permitting photosynthesis at the ice-water interface (Hancke et al., 2018) could have triggered the productivity of IPSO$_{25}$ source diatoms, even if sea ice extent would have been lower. Thinner ice and accelerated melting during spring may have resulted in a largely ice-free sea surface during summer promoting phytoplankton (biomarker) productivity. In addition, increased melting of sea ice could have contributed to higher primary production by releasing nutrients in a stabilized water column. This could have led to a higher deposition of sea ice diatoms and IPSO$_{25}$ by sinking of these phytoplankton blooms during spring and summer (Palanques et al., 2002). The declining mSSIC and mSSIT (supplement S5) support the interpretation of sea ice thinning. Therefore, we suggest that increasing concentrations of both IPSO$_{25}$ and phytoplankton-derived biomarker lipids accordingly reflect more pronounced ice-edge conditions and/or a distinct seasonality in spring and summer conditions along the WAP over the past 240 years.

As the distribution of IPSO$_{25}$ is sensitive to local oceanographic conditions (Smik et al., 2016a), biomarker-based sea ice studies require an interpretation that takes the specific environmental characteristics of the region into account. In this context, we generally expect influences of meltwater input from glacial melting during summer (Meredith et al., 2018), additional nutrient input from the APCC and intense mixing at the Peninsula Front. We suggest that high fluctuations in sea ice cover, sea ice thickness and water temperature may stimulate phytoplankton growth rather than stable conditions with very high and long lasting or low ice cover and/or ice-free sea surface (e.g. Xiao et al., 2013). We hence strengthen the need to compare the individual concentration records of IPSO$_{25}$ and phytoplankton biomarkers rather than using the IPSO$_{25}$ (and PIPSO$_{25}$) record alone to deduce sea ice conditions (see also Müller et al., 2011, 2012).

We further consider records of MSA, an organic aerosol archived within ice cores, which is associated with marine biological activity during sea ice breakup and which is used as a proxy for sea ice reconstructions. Influenced by timing, duration and spatial extent of sea ice breakup, MSA concentrations are linked with winter sea ice extent in some regions and summer productivity within the sea ice zone in other regions of Antarctica (Thomas et al., 2019 and references therein). Here we use records of MSA from the Dyer Plateau on the AP as well as a stacked MSA

record based on three regional ice cores (James Ross Island, Dyer Plateau, and Beethoven Peninsula) (Abram et al., 2010) that reflect winter sea ice dynamics in the Bellingshausen Sea. Both records display an overall decreasing trend in MSA concentrations since 1900 CE indicating less sea ice (Fig. 5). Another record from the Bruce Plateau ice core shows the annual net accumulation (Porter et al., 2016). The increase of snow accumulation is suggested

to be linked to the sea ice extent in the Bellingshausen Sea and indicates a distinct decrease of sea ice in the second half of the 20th century which agrees with the MSA and modelled data. All ice core records show some agreement with the mSSIC for the East Bransfield Basin (PS97/072-2) but are opposite to our biomarker records and sea ice indices for all three core sites. This is likely due to the fact that our sediment records reflect local to regional changes strongly influenced by the AP as a geographic barrier and the complex oceanography within the Bransfield

Strait from interaction of BSW and WSW. As both the Dyer Plateau and the stacked MSA records are dominated by large-scale winter sea ice cover variability in the Bellingshausen Sea (centered between 70° and 100°W) (Abram et al., 2010), we suggest that the regional sea ice variability within the Bransfield Strait archived in our sediment cores is not well reflected in the ice core records.

Additionally, we took the latitudinal movement of the spring sea ice edge from modelled data (mSSIE, Fig. 5) into

account, which displays a southward shift down to 63.5°S reflecting sea ice retreat and proposes the occasional absence of spring sea ice at all core sites since the 1970s. The spatial shift of the sea ice edge must be treated with caution because the model does not resolve regional impacts, coastal and peninsula interactions and seasonal input of drift ice from the Weddell Sea. The MSA-based winter sea ice edge (WSIE, Fig. 5) (Abram et al., 2010) displays the same decreasing trend in the Bellingshausen Sea but is located 3° to the south of the modelled ice edge (from

65° to 66°S). The fact that our core sites are located north of this projected WSIE shift is another argument why the ice core MSA cannot be considered to reflect sea ice conditions in our study area, which, according to the ice core data would have been free of sea ice during the entire 20th century.

We relate the divergence of ice core and modelled sea ice data from our sediment core data firstly, to the different spatial coverage and geographic origin of the environmental signals archived within the ice cores and, secondly,

the aspect that AWI-ESM2 cannot resolve the AP sub-aerial and marine topography and have difficulties in capturing local to regional near coastal sea-ice dynamics in the study region. In fact, that our sediment records reflect local to regional impact of the BSW and WSW that carry opposite sea ice and water mass properties and neither represent sea ice properties of the Bellingshausen Sea nor the Weddell Sea. In addition, as the AP is acting as a geographic barrier between these water masses, the region is highly sensitive to oceanographic variabilities

driven by atmospheric patterns. For example, it is suggested that strong westerly winds and a positive SAM

diminish the inflow of WSW into the Bransfield Strait (Dotto et al., 2016). Although the WAP is studied quite well the lack of high resolution records that display local sea ice calls for further sea ice paleo record surveys.

### 4.4 Comparison of marine temperature proxy records with model and ice core data

Comparison of GDGT-based temperatures with modelled subsurface ocean temperature mSOT reveals a general disagreement over the 20[th] century (Fig. 6). Only at the Orleans Trough (PS97/068-2) high $SOT^{TEX}$ might reflect an impact of atmospheric temperatures but as all other temperature estimations are much lower we assume that enhanced water column mixing at the Peninsula Front might diminish this effect (see section 4.1). During the 19[th] century, $SOT^{TEX}$-based cold (around 1850s and 1900 CE) and warm events (from 1860 to 1880 CE, and around 1910 CE), respectively, agree better with mSOT at all core sites than in the 20[th] century. $SOT^{OH}$ does not correspond to mSOT except since the 1990s when both data sets reflect the modern warming. SSST from diatoms show a short cool period around 1900 CE similar to $SOT^{TEX}$ and modelled data. In general, biomarker derived temperatures point to a slight cooling trend over the last 240 years at the WAP which fits to the observed deep water cooling between 1960 and 1990 (Ruiz Barlett et al., 2018). Simultaneously, this cooling contradicts with our mSOT and observations of surface water warming that seem to be linked to the increasing positive SAM since the mid-20[th] century (Abram et al., 2014; Ruiz Barlett et al., 2018).

As our model includes a transient greenhouse gas forcing the highly variable but continuously increasing mSOT (and mSAT) matches the observed trends in atmospheric warming derived from stable isotope ice core and meteorological data (Fig. 6). The ice core records of $\delta^{18}O$ records at Bruce Plateau (Goodwin et al., 2016) and $\delta D$ records from James Ross Island (Abram et al., 2013) display the large-scale air temperature rise in the sector of the Bellingshausen Sea and the Antarctic Peninsula region. The same upward trend is seen in mean surface air temperatures from meteorological stations on the WAP parallel to rising global atmospheric carbon dioxide concentrations (Fig. 6). However, we note that ice cores represent a large regional scale and meteorological station records are influenced by e.g. altitude, morphology and local wind patterns, while GDGT-based derived ocean temperatures depict a local to regional subsurface marine record controlled by BSW and WSW. We also note that water mass transformation during sea ice cover is possible due to formation of cold, salty brine waters and enhanced vertical mixing with cold water masses occur (Abernathey et al., 2016). Further, sea ice melting in spring enhances the stratification of the upper water column, restricts heat exchange between the subsurface ocean and atmosphere and could lead to a cold bias in subsurface temperature biomarker records. Nevertheless, the abrupt warming in our ocean temperature records in the 1990s follows the significant and extraordinary warming shown in air temperature records accompanied by the steep rise in atmospheric carbon dioxide.

### 4.5    Sea ice evolution and large-scale atmospheric circulation patterns

While sediment records integrate environmental conditions of several years, the influence of highly seasonal climate modes or events may not be properly dissolved which could explain the weaker relationships. However, since atmospheric circulation affects the heat and sea ice distribution along the WAP especially during spring time (Clem et al., 2016), we expect patterns of ENSO and/or SAM to leave a footprint in our spring sea ice IPSO$_{25}$ record. Several studies suggest an enhanced influence of ENSO and SAM on Antarctic temperatures with increasing greenhouse gas concentrations, so their relation to sea ice is a crucial factor for sea ice and climate predictions (Rahaman et al., 2019; Stammerjohn et al., 2008b). For example, the atmosphere-ocean-sea ice interactions impact the WAP strongly through increased northerly winds during an in-phase -ENSO/+SAM mode. They lead to a strong, positive feedback of atmospheric warming amplification due to shorter sea ice seasons, thinner sea ice cover with more leads permitting an enhanced heat flux from the ocean (Stammerjohn et al., 2008a). Further, +SAM increases the presence of BSW and CDW in the Bransfield Strait and reduces the influence of WSW resulting in higher SSTs in the Bransfield Strait (Dotto et al., 2016; Ruiz Barlett et al., 2018).

We compare IPSO$_{25}$ from all core sites with a tree-ring based ENSO reconstruction (Li et al., 2013) and SAM data from proxy records including the full mid-latitude to polar domain of the Drake Passage (Abram et al., 2014) (Fig. 7). Both, ENSO and SAM have oscillating positive and negative periods and SAM shows a slight decrease until 1860 CE. Since 1930 CE, SAM, and since 1960 CE, ENSO, increase again and reach maximum positive states in the 2000s. When comparing biomarker and circulation patterns, SAM matches best with elevated HBI concentrations, especially at the coastal core sites, relating a higher accumulation of IPSO$_{25}$ to a +SAM. During a +SAM, stronger westerly winds cause a southward shift of the low-pressure cell over the Bellingshausen Sea and the strengthening of the polar frontal jet (Marshall et al., 2006). At this time, temperature anomalies along the WAP are very small and not even detectable at e.g. the southwest Vernadsky/Faraday Station (Marshall et al., 2006) but a remarkable "Föhn" effect (Klemp and Lilly, 1975) leads to rising summer air temperatures on the eastern AP leeside. Nevertheless, our records suggest that a +SAM is positively related to the content of IPSO$_{25}$ and HBI Z-triene in Bransfield Strait sediments, especially since the mid-20[th] century. Although a +SAM is associated with higher SSTs due to a higher presence of BSW in the Bransfield Strait and reduced WSW (Dotto et al., 2016; Ruiz Barlett et al., 2018) we do not see any relations to our temperature proxies (supplement S6). We suggest that a higher presence of sea ice (indicated by higher IPSO$_{25}$ concentrations) could lead to lower annual mean ocean temperatures because brine rejection during sea ice formation leads to cold water formation in the Bransfield Strait (Abernathey et al., 2016).

The pattern of ENSO is not or even negatively related with biomarker concentrations in the 19th century (especially at core site PS97/072-2) and more positively in the 20th century. The recent shift to a positive ENSO is accompanied by increased IPSO$_{25}$ concentrations. After Yuan (2004) a +ENSO causes sea ice advance under cold conditions in the Weddell Sea and the Bransfield Strait, and warm, moist conditions in the Southern Pacific Ocean.

However, due to observations of recently rising atmospheric temperature (Stastna, 2010), ocean temperature (Cook et al., 2016) and declining sea ice cover along the southern WAP, a +ENSO seems to be more likely related to warm and sea ice reduced conditions along the WAP in the studied period. Nevertheless, we observe that the IPSO$_{25}$ production at the coastal core sites (PS97/056-1 and 068-1) corresponds to +ENSO since the 1980s. Neither SAM nor ENSO alone seem to exert a consistent control on IPSO$_{25}$, phytoplankton production or ocean

temperature in the Bransfield Strait (supplement S6). A +ENSO together with +SAM seem to be linked to higher IPSO$_{25}$ concentrations especially in the 20th century, which agrees with previous suggestions regarding the impact of atmospheric circulation patterns on sea ice conditions (Barbara et al., 2013; Etourneau et al., 2013) but as meltwater discharge seems to impact environmental records at our near-coastal cores sites this suggestion remains hypothetical.

**4.6    Interpretation of combined paleoenvironmental biomarkers**

While all core sites exhibit increasing concentrations of both open marine and sea ice biomarkers we simultaneously observe an ocean cooling (mainly indicated by SOT$^{OH}$). We assume subsurface ocean cooling to be linked to the release of cold, salty waters during sea ice formation (Abernathey et al., 2016) and enhanced vertical mixing in winter (Frew et al., 2019) resulting from an advance of sea ice cover in the 20th century. Instead

of a master stratigraphy for all three cores we defined site-specific paleoenvironmental units characterized by low, moderate and high spring sea ice cover estimates (see vertical bars Fig. 3). Due to site-specific oceanographic characteristics, the temporal duration of the defined units may differ between the core sites. This sub-division is of qualitative nature and does not address quantified sea ice cover. It is mainly based on sea ice proxies (IPSO$_{25}$, PIPSO$_{25}$, WSI and *F.curta*) but also considers subsurface ocean temperatures.

*Unit A:  Low sea ice cover and high ocean temperatures:* In unit A (orange bar, Fig. 3) sea ice proxies of IPSO$_{25}$, PIPSO$_{25}$ as well as WSI and *F.curta* are mostly below average and mark low winter and spring sea ice cover accompanied by relatively high ocean temperatures (SOT$^{OH}$). The sea ice minimum and probably ice-free summers occur in the mid-19th century with a remarkable WSI low at Orleans Trough (PS97/068-2) and low WSI with high HBI triene values at the East Bransfield Basin (PS97/072-2). The reduced sea

ice cover could explain the relatively high subsurface ocean temperatures as a result of diminished vertical mixing (Frew et al., 2019) and enhanced input of BSW. As reason that unit A only extends to the end of

the 19th century at the two northernmost core sites while it prolongs to the 1920s at the Trinity Island site we assume that the influence of BSW at the Orleans Trough and East Bransfield Basin decreases towards the 20th century. Also, a rising wind driven sea ice transport from the Weddell Sea (Holland and Kwok, 2012) could possibly end unit A earlier at the northernmost core sites due to a weaker westerly wind belt (Koffman et al., 2014) as well as a shift from +SAM to -SAM (Abram et al., 2014) during the 19th century.

*Unit B:* *Moderate winter and spring sea ice cover with decreasing temperatures.* In unit B (green bar, Fig. 3) three of the four sea ice proxies are above average and show an increase in sea ice for both winter and spring at all core sites in accordance with decreasing ocean temperatures. Especially at Orleans Trough (PS97/068-2) high short-term fluctuations in sea ice seem to be common, which may relate to the interplay of water masses and a shift of the Peninsula Front, influenced by WSW in the southeast and BSW in the northwest (Fig. 1). The advance of sea ice appears about 30 years earlier at the East Bransfield Basin (PS97/072-2) than at Trinity Island (PS97/056-1) and induces a transformation towards colder subsurface ocean temperatures through brine rejection and regular sea ice melt in spring and summer (Abernathey et al., 2016). We suggest that a northward shift of the westerly wind belt with a weakening of the local winds (Koffman et al., 2014) and -SAM (Abram et al., 2014) could have increased the influence of colder, ice-rich WSW (Dotto et al., 2016) and lead to a propagation of moderate sea ice cover from northeast to southwest over several decades. The development towards moderate sea ice cover at Trinity Island since the 1920s contrasts suggestions from Barbara et al. (2013) who interpreted near-coastal diatom assemblages and HBIs at the southern WAP to reflect decreasing sea ice cover and warmer SSTs. Therefore, we assume that the Peninsula Front might have extended southwards due to decreased westerly winds and Trinity Island was temporary dominated by WSW.

*Unit C:* *High but variable sea ice cover and low ocean temperature.* In unit C (blue bar, Fig. 3) $IPSO_{25}$, $PIPSO_{25}$, WSI and *F.curta* clearly indicate the highest sea ice cover over the studied period. The majority of ocean temperature proxies shows a further decrease towards low values between the 1970s and 1990s at all sites indicating a shift towards cold, salty waters due to brine rejection from sea ice (Abernathey et al., 2016; Frew et al., 2019). In general, the high variability in sea ice cover seen in our study area matches other marine sediment records near Anvers Island (Barbara et al., 2013). These high seasonal contrasts promoted enhanced open marine biomarker production (Gonçalves-Araujo et al., 2015), evident in higher concentrations of $IPSO_{25}$ and HBI trienes (Fig. 3, supplement S4) and sedimentation rates (Fig. 2), that are fueled by high nutrient release through sea ice melting (Vernet et al., 2008). All core sites experience a sudden sea ice minimum in both winter and spring and a sudden temperature rise in the 1960s at Orleans

Trough (PS97/068-2), the 1970s at Trinity Island (PS97/056-1) and the 1980s at the East Bransfield Basin (PS97/072-2). This might be linked to a growing input of BSW since 1960 when SAM changes from negative to positive (Fig. 7, Abram et al., 2014) and strengthening of the westerly winds (Bracegirdle et al., 2018). This is contrasted by a rapid advance of sea ice since the 1970s at the coastal core sites (PS97/056-1 and 68-2) that is likely driven by enhanced freshwater pulses from retreating glaciers at the AP (Cook et al., 2005; Kunz et al., 2012) which do not affect the remote East Bransfield Basin core site (PS97/072-2).

*Unit D: High sea ice cover and warm ocean temperatures.* The last unit D (purple bar, Fig. 3) is characterized by a high sea ice cover and rapidly rising ocean temperatures. The onset of this trend towards a warm subsurface ocean (Cook et al., 2016) is present at all core locations since the mid-1990s. Sea ice cover tends to increase towards maximum values in the 2000s and seems to reflect recent observations of sea ice cover rebounds in the Bellingshausen Sea and the WAP after 2005 CE (Hobbs et al., 2016; Schofield et al., 2018). At the same time the high ocean temperatures speak against cold water formation from sea ice which let us assume that the inflow of warm BSW is exceptionally high and dominates the temperature proxies during a strong SAM. Further, increased meltwater discharge could support sea ice formation from freshwater (Haid et al., 2017). Near Trinity Island (PS97/056-1) the drastic low of WSI and *F.curta* might show the higher sensitivity of diatoms to short term changes compared to HBIs. Since the last unit is very short, the interpretation of warm ocean temperature together with a high sea ice cover is rather tentative. In contrast to units A to C this unit occurs simultaneously at all core sites and marks an environmental change on a larger scale that overprints the former strong local impacts.

From the temporal onset of different environmental states in the Bransfield Strait we interpret that sea ice and ocean temperature distributions are mainly the result of the interplay between BWS and WSW that is controlled by westerly winds and SAM. The influence of these water masses was potentially overprinted by meltwater discharge at the coastal core sites in the second half of the 20th century. Figure 8a illustrates how a mainly +SAM and the dominance of BSW lead to the low sea ice cover in the Bransfield Strait describing unit A. In unit B, a shift from +SAM to -SAM in the first half of the 20th century supports a sea ice advance first at the core sites that are closest to the WSW (PS97/68-2 and 72-2) and later at the southernmost core site (PS97/056-1) (Fig. 8b). With a +SAM in the 1960s a short-term sea ice minimum appears from the southwest to the northeast indicating the growing dominance of BSW (Fig. 8c). Despite a shift towards +SAM and the rising dominance of BSW in the following decades, sea ice advance and low ocean temperatures occur at the coastal core sites defining unit C. We assume this to be a strong imprint of meltwater input which promotes sea ice formation because of a higher freezing

point. The meltwater originates from fast retreating glacier fronts and glacial melting since the 1980s (Cook et al.,

2005) due to rising greenhouse gases and atmospheric temperatures (Fig. 6). The rise in ocean temperatures in unit

D might relate to the establishment of +SAM and continuous input from BSW (Fig. 8d). Since the 2000s, glacial

melting increased significantly (Rignot et al., 2019) and this enhanced freshwater input may have increased the

sea ice extent. It is also likely that biomarker production in both sea ice and open marine environments benefits

from a sudden sea ice retreat (as it was recorded by satellites for 2000 to 2005, Fig. 4) due to thinner sea ice

(Hancke et al., 2018) and a higher nutrient overturn by sea ice melting (Vernet et al., 2008).

From our reconstructions of sea ice and ocean temperatures we argue that SAM was the main driver of water mass

distribution in the Bransfield Strait in the past 240 years as it was also found in studies covering the last 40 years

(Clem et al., 2016; Dotto et al., 2016; Stammerjohn et al., 2008b; Yuan, 2004) but local input from meltwater is

able to increase near-coastal sea ice formation (and related primary production) during sea ice diminishing

conditions which can overprint the dominating climate mode. This may explain why the positive relation between

+SAM and IPSO$_{25}$ concentrations are highest at the near-coastal core sites, although +SAM would lead to a higher

BSW inflow and a reduction of sea ice.

## 5    Summary and Conclusions

We analyzed the spring sea ice biomarker IPSO$_{25}$ and other biomarkers as well as diatom assemblages in three

sediment cores from the Bransfield Strait along covering the past 240 years and combined our results with

numerical model data, satellite observations, temperature records and paleo records of atmospheric circulation

patterns. We note that the interpretation of the biomarker data for past sea ice estimates in Antarctica is strongly

impacted by the origin of water masses and mixing, nutrient input and dynamics of sea ice-related primary

production. While IPSO$_{25}$ concentrations agree with satellite sea ice data, they seem to contradict the long-term

large-scale ice core and model data. We note that the significance of our coupled climate model is limited due to

the complex oceanography in the Bransfield Strait that is controlled by atmospheric circulations patterns and

forced by enhanced glacial melting during the past 50 years. We also emphasize the fact that local coastal

influences, high sea ice dynamics and thinner sea ice promoting the production of both sea ice diatoms and open

marine phytoplankton may affect the interpretation of IPSO$_{25}$ and the sea ice index PIPSO$_{25}$. When estimating

spring sea ice cover, the strong sensitivity of IPSO$_{25}$ to local influences such as water masses, coastal interaction

and, e.g. a higher sea ice algae productivity resulting from thinner ice cover need to be taken into account. We

hence recommend to consider additional phytoplankton data instead of constructing sea ice estimates on IPSO$_{25}$

and PIPSO$_{25}$ records solely.

Although ENSO and SAM are postulated to influence sea ice and heat distribution in the Bransfield Strait our

biomarker data does not show clear relationship between the long-term ocean temperature development and these

patterns has been suggested from recent studies. However, ENSO and/or SAM both seem to affect the sea ice

regime in the Bransfield Strait. Based on sea ice biomarkers and sea ice indices, we roughly divided the 240-year

records into four environmental units with different timings (Fig. 8):

Unit A    *Low sea ice cover and high ocean temperatures* due to a dominance of warm BSW during a +SAM in

the 19$^{th}$ century (Fig. 8a).

Unit B    *Moderate winter and spring sea ice cover with decreasing temperatures* occur when +SAM shifts

                towards -SAM and the input of colder and sea-ice rich WSW in the Bransfield Strait increases (Fig. 8b).

Unit C    *High but variable sea ice cover and low ocean temperature* establish over several decades propagating

                from the northeast to the southwest with a change from -SAM to +SAM and diverging sea ice conditions

between the offshore and coastal core sites due to glacial melting and meltwater input (Fig. 8c).

Unit D    *High sea ice cover and warm ocean temperatures* are related to a peak of +SAM and BSW input during

                a short interruption of atmospheric warming and glacial melting (Fig. d).

We conclude that the different timing of the units mirrors the decadal change of dominating water masses at every

core site. A dominance of +SAM promoted the enhanced input of BSW in the Bransfield Strait that lead to a sea

ice reduction at the remote core site (unit C, Fig. 8c). In contradiction to that sea ice cover was enhanced at the

near-coastal core sites due to a strong overprint from enhanced meltwater input from glaciers since the 1960s and

ocean cooling resulted from cold brine rejections during sea ice formations.

**Data Availability**

All data are available at the open access repository www.pangaea.de
(https://doi.pangaea.de/10.1594/PANGAEA.918808).

**Author contributions**

The study was conceived by MV and JM. Data collections and experimental investigations were done by MV together with PC, LR, PM and CBL (sampling, diatoms, dating), WG (dating), OE (diatom transfer functions), JM and GM (HBIs, GDGTs), XS and GL (modelling and supplement Fig. S3), CH (satellite sea ice data), and TO (ice cores). MV drafted the manuscript and figures. JM supervised the study. All authors contributed to the interpretation and discussion of the results and the conclusion of this study.

**Competing interests**

None of the authors has a conflict of interest.

**Acknowledgement**

We thank the captain, crew and chief scientist Frank Lamy of RV Polarstern cruise PS97. Denise Diekstall, Jens Hefter, Ingrid Stimac and Ruth Cordelair are thanked for their laboratory support. We also thank Andrés Cádiz for the help on diatom slide preparations and counts on core PS97/056-1. Simon Belt is acknowledged for providing the 7-HND internal standard for HBI quantification. Financial support was provided through the Helmholtz Research grant VH-NG-1101 and the Helmholtz Excellence Network "The Polar System and its Effects on the Ocean Floor" ExNet-0001. Partial financial support from the Research Center Dynamics of High Latitude Marine Ecosystems (FONDAP-IDEAL 15150003, Chile) and the Center for Oceanographic Research COPAS Sur-Austral (AFB170006, Chile) is acknowledged. We thank two anonymous reviewers for their constructive comments that helped us to improve the manuscript.

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

**Figures**

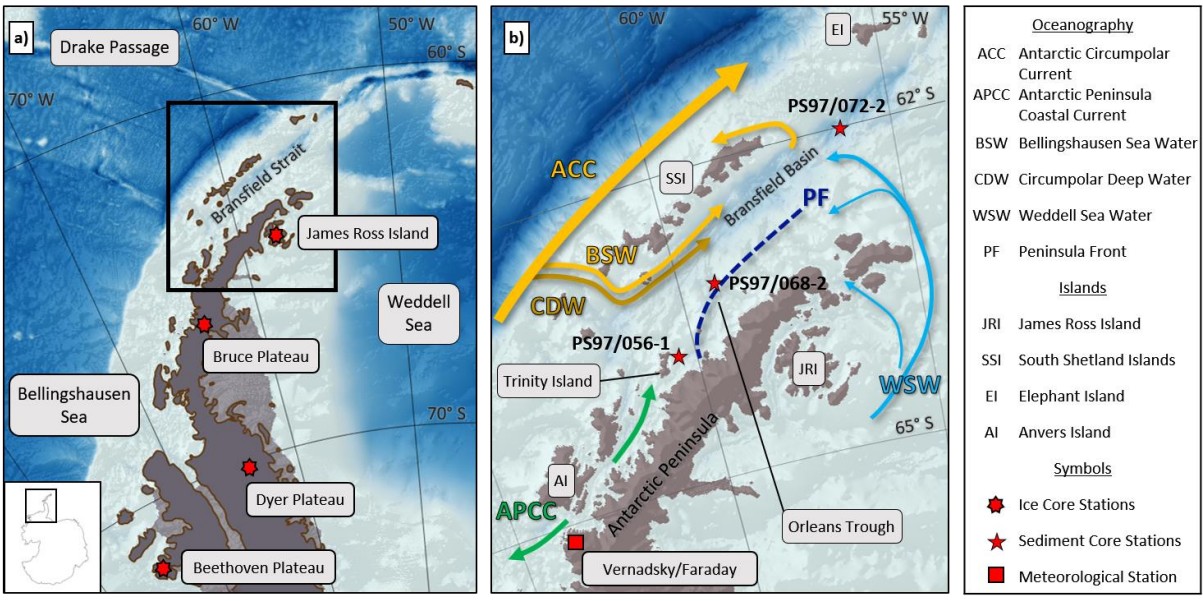

**Figure 1: a) Overview map of the Antarctic Peninsula with the position of the ice cores from Dyer Plateau, Beethoven Plateau and James Ross Island (Abram et al., 2010), Bruce Plateau (Goodwin et al., 2016) and bathymetric features in the Bellingshausen Sea, the Weddell Sea and the Drake Passage. B) Oceanographic setting in the study area (modified after Hofmann et al., 1996; Moffat and Meredith, 2018; Sangrà et al., 2011), sediment and ice core sites, and geographic locations mentioned in the text. Maps were generated with QGIS 3.0 (2018) and the bathymetry was taken from GEBCO_14 from 2015.**

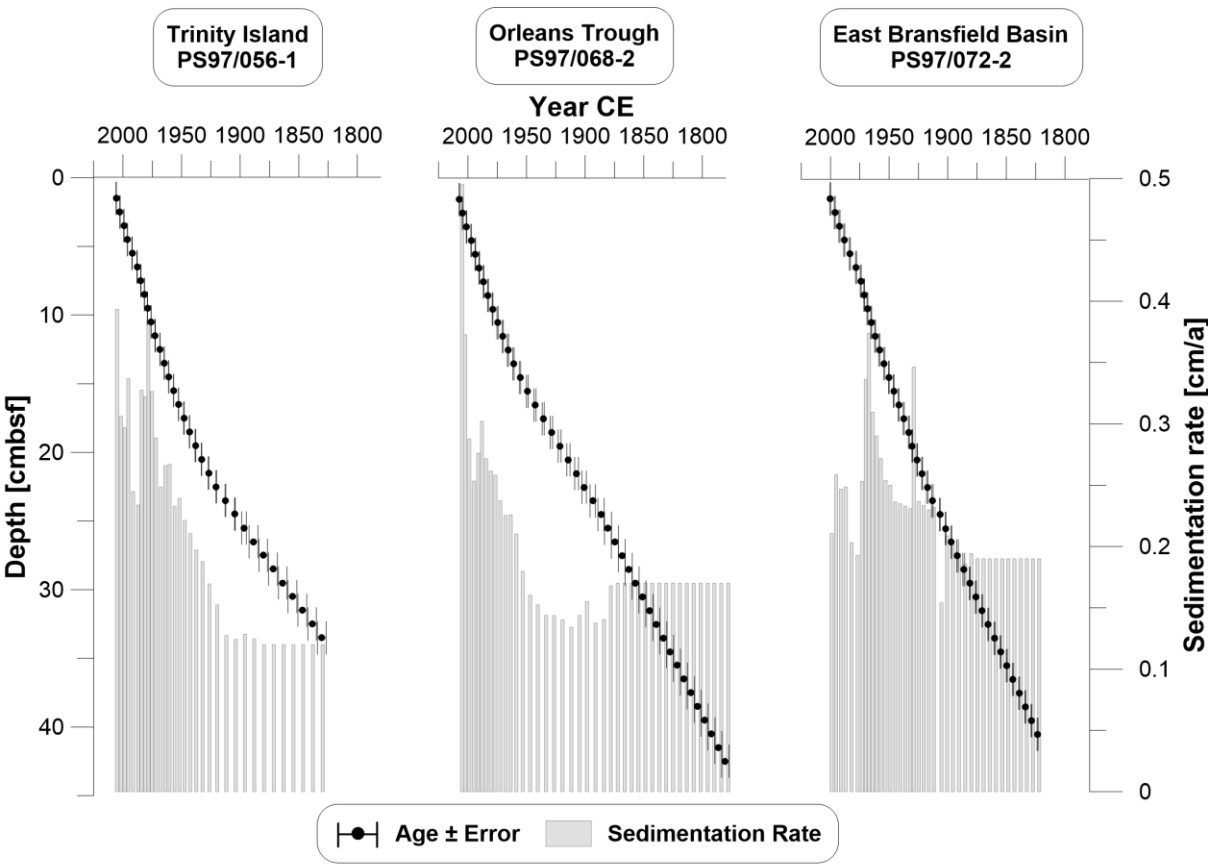

**Figure 2: Age-depth models with error bars of all three cores. The sedimentation rate is displayed in grey bars. Ages are based on radiometric dating and were extrapolated prior to 1880 CE for all cores based on their average respective sedimentation rate of the three lowermost, reliable dated intervals. All plots were done with Grapher™ 13.**

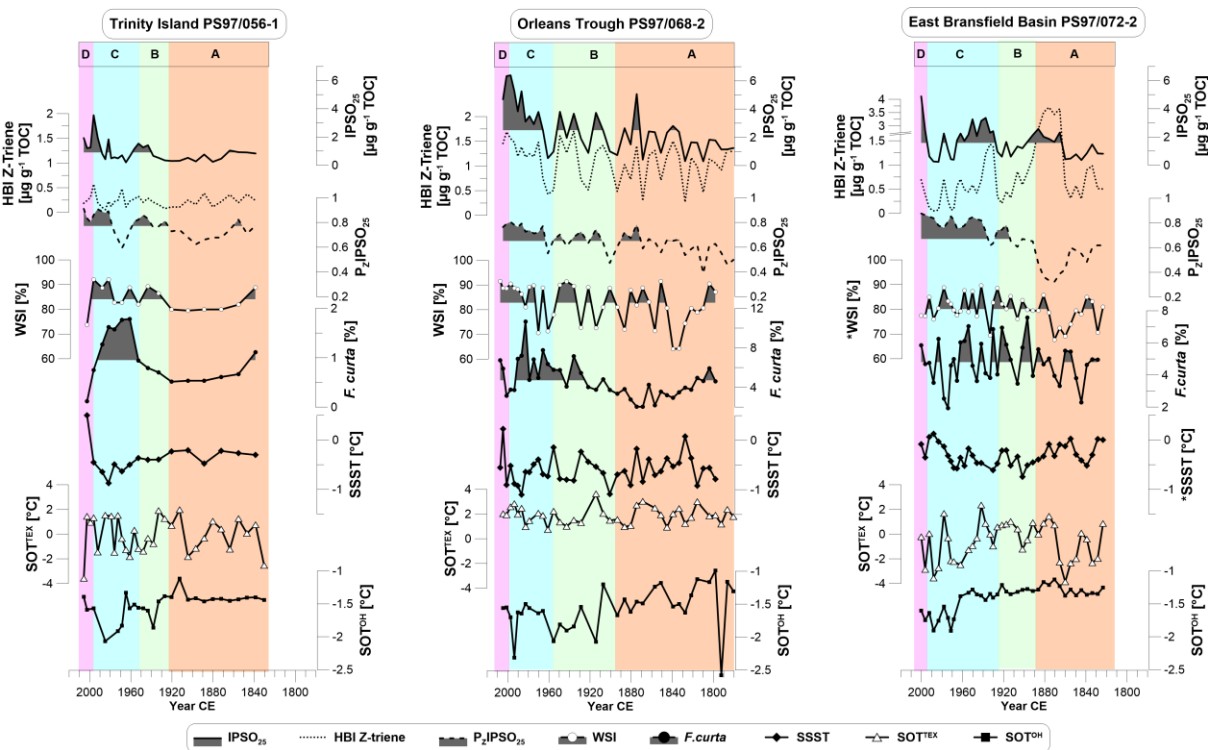

**Figure 3: Biomarker composition of the three sediment cores showing concentrations of (from top to bottom) IPSO$_{25}$ and HBI Z-trienes, the sea ice index P$_Z$IPSO$_{25}$, diatom-derived winter sea ice (WSI) concentrations and temperatures of summer sea surface temperatures (SSST), subsurface ocean temperature derived from TEX$^L_{86}$ (SOT$^{TEX}$), and OH-GDGTs (SOT$^{OH}$). Data marked with * are from the trigger core PS97/072-1. Vertical colored bars denote the environmental units A to D described in section 4.5. The shadings mark values above the mean of each biomarker in the respective core.**

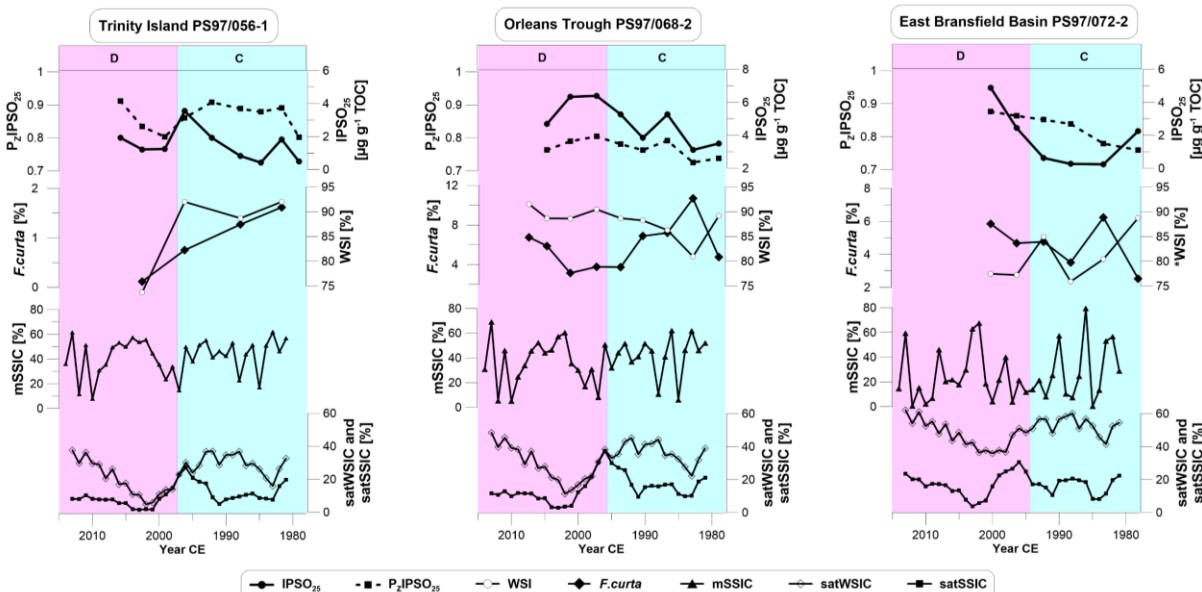

**Figure 4: Concentrations of (from top to bottom) IPSO25, PzIPSO25, WSI compared to modelled spring sea ice concentrations (mSSIC) and satellite derived winter and spring sea ice concentrations (satWSIC and satSSIC, 5 year running mean) from the National Snow and Ice Data Center (NSIDC, Cavalieri et al., 1996) for all three core sites. Data marked with \* are from the trigger core PS97/072-1. Vertical bars denote the environmental units C and D.**

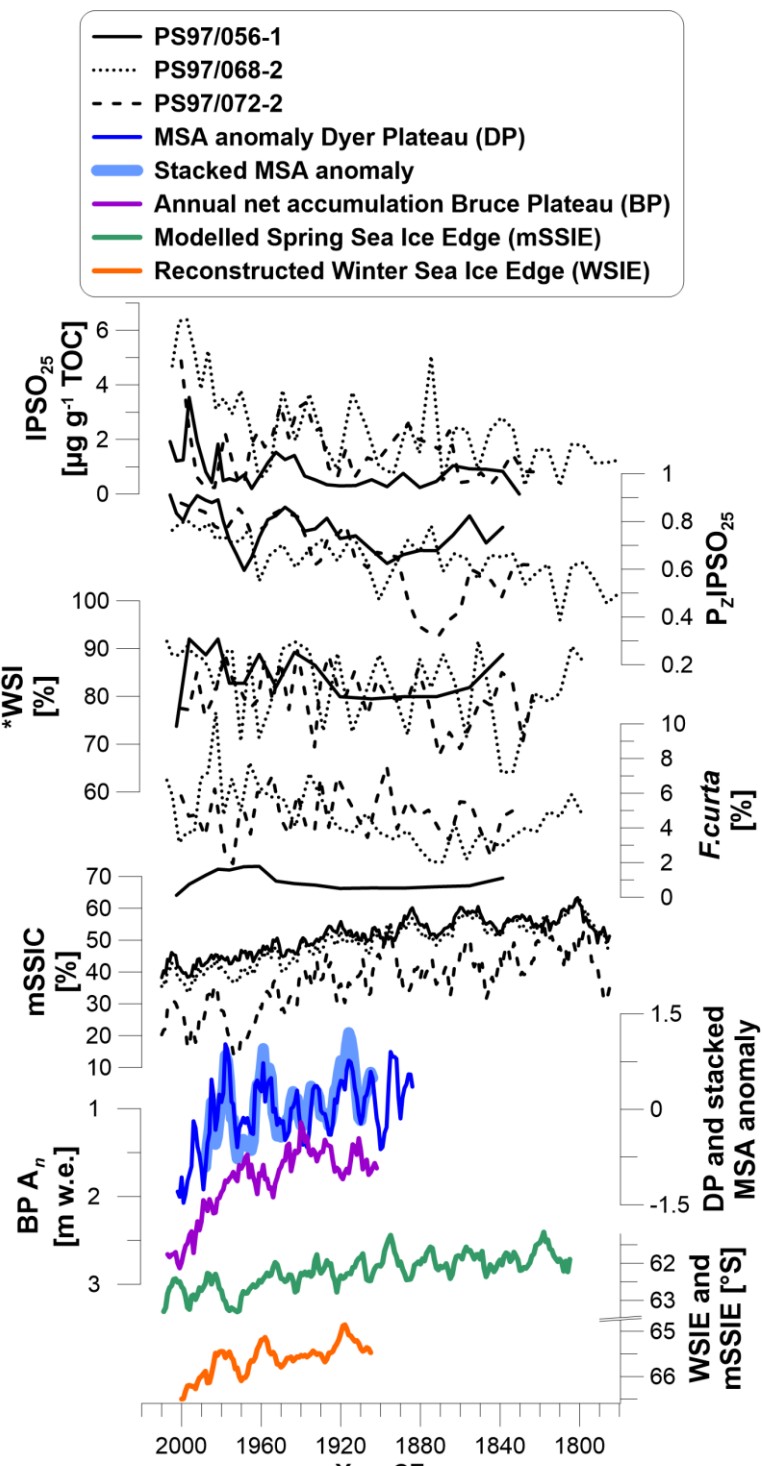

**Figure 5: The biomarker (from top to bottom) IPSO$_{25}$, sea ice index P$_Z$IPSO$_{25}$, and winter sea ice concentration (WSI) from diatom assemblages compared to modelled spring sea ice cover (mSSIC, 10 year running mean), MSA anomaly from Dyer Plateau and stacked MSA covering the Bellingshausen Sea sector (5 year running mean, Abram et al., 2010), annual net accumulation from Bruce Plateau (5 year running mean, Goodwin et al., 2016), modelled spring sea ice edge latitude (mSSIE, 10 year running mean) and reconstructed winter sea ice edge latitude from MSA (WSIE, 10 year running mean, Abram et al., 2010). Data marked with * are from the trigger core PS97/072-1.**

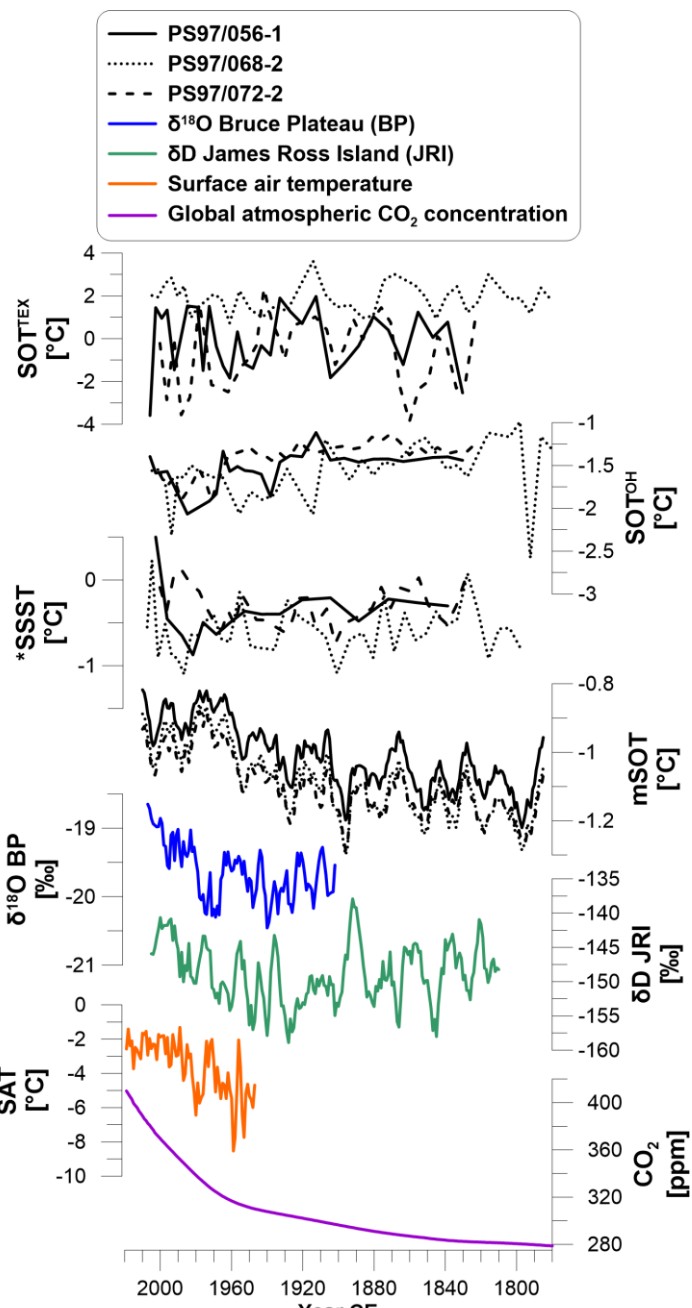

**Figure 6: Biomarker derived subsurface ocean temperatures based on TEX$^L_{86}$ (SOT$^{TEX}$), and hydroxylated GDGTs (SOT$^{OH}$), and summer sea surface temperatures (SSST) derived from diatom assemblages compared to modelled subsurface ocean temperature (mSOT), stable isotope ice core records from the Bruce Plateau (BP, δ$^{18}$O, 5 year running mean; Goodwin et al., 2016) and from James Ross Island (JRI, δD, 5 year running mean; Abram et al., 2013). Annual means surface air temperature (SAT) is derived from four meteorological stations (annual mean from stations O'Higgins, Faraday/Vernadsky, Bellingshausen and Jubaney, British Antarctic Survey, 2013) and global annual atmospheric carbon dioxide concentrations are combined from Meinshausen et al. (2017) and Tans and Keeling (2020). Data marked with * are from the trigger core PS97/072-1.**

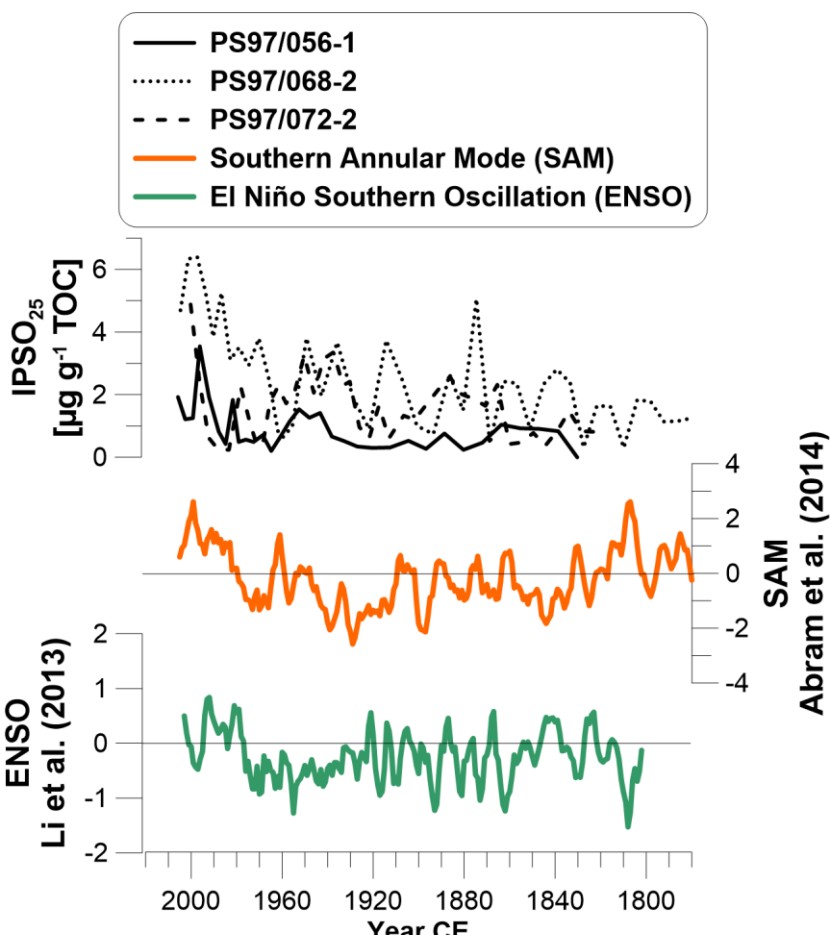

**Figure 7: Concentrations of biomarker IPSO$_{25}$ in all three sediment cores compared to circulation pattern of the Southern Annular Mode (SAM, 5 year running mean; Abram et al., 2014), and the El Niño Southern Oscillation (ENSO, 5 year running mean; Li et al., 2013).**

**19th century:**
- dominance BSW
- low sea ice cover and high ocean temperatures

**1st half 20th century:**
- propagating WSW
- sea ice advance and ocean cooling

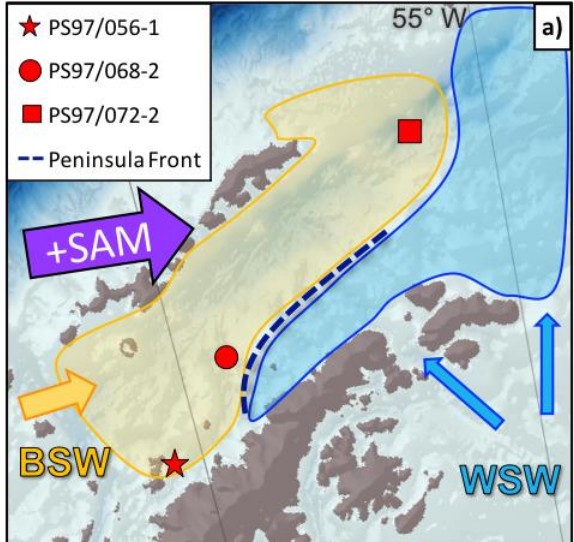
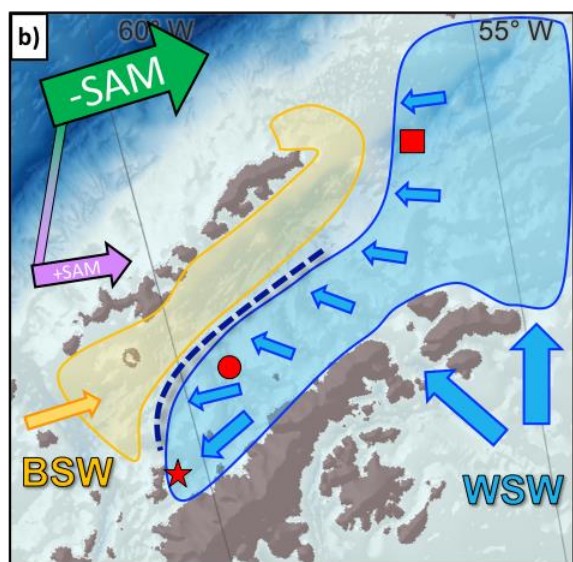

**2nd half 20th century:**
- propagating BSW and high meltwater input
- high sea ice cover and low ocean temperatures

**After 2000:**
- dominance BSW and diminished meltwater input
- high sea ice cover and high ocean temperatures

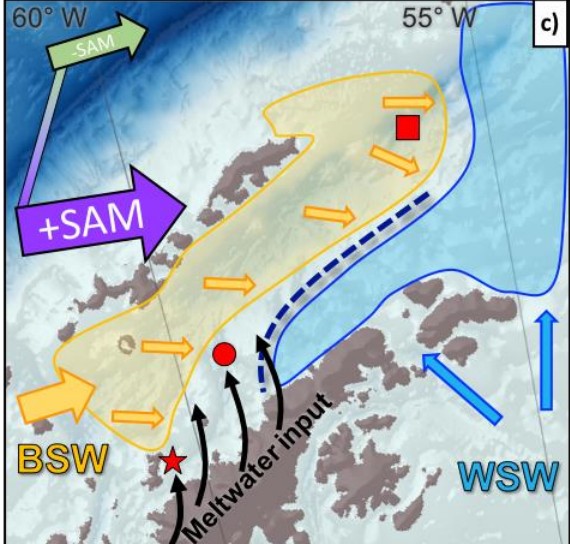
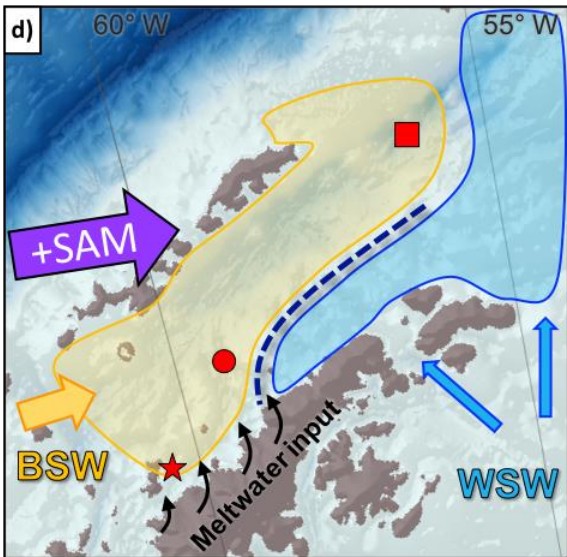

**Figure 8: The illustration of the four environmental units and their dominant drivers of a) low sea ice cover and high ocean temperatures from dominating BSW described in Unit A, b) moderate winter and spring sea ice cover with decreasing temperatures that propagate from the Weddell Sea to the southern WAP in Unit B, c) high but variable sea ice cover and ocean temperature lows under advancing BSW and +SAM and additional meltwater pulses of Unit C, and d) high sea ice cover under a warm ocean temperature reversal as a result of BSW dominance and meltwater input from the AP in Unit D. Thick and long arrows indicate a strong forcing, thin and short arrows a weak forcing. The arrows of SAM in unit B and C describe the transition of the mode from the smaller to the bigger arrow.**