# Peer review of "Sea Ice dynamics in the Bransfield Strait, Antarctic Peninsula, during the past 240 years: a multi-proxy intercomparison study"

_Climate of the Past, 2020_

## Referee Comment (RC1) · Anonymous Referee #1 · 19 Jun 2020

Given the dramatic climatic and environmental impact of the ongoing global warming on the Antarctic Peninsula, numerous studies have scrutinised and examined the modern variability of sea ice over the past few years. However, we still know a little about its pre-industrial evolution, i.e. under natural climatic forcing, and its evolution throughout the transition toward an increasingly "industrialised world". To this aim, Vorrath et al. present here a very interesting study about sea ice dynamic in the Northern Antarctic Peninsula during the past 200 years. They combine molecular and micropaleontological proxy records from three strategically located and well dated marine sediment core sites located within the Bransfield Strait, NW of the Peninsula. After comparing their records with satellite, ice core and model data, the authors document changes in sea

ice in the NW of the Peninsula since the 1800's and the potential processes (ENSO and SAM) controlling its recent past variability.

This manuscript is well written, concise, clear, includes adequate references and addresses most of the critical questions concerning the proxy used and their interpretation. The authors also introduce all the potential biases and limits of their records.

This study is of major and broad interest for both the paleoclimate and paleoceanography communities but also for oceanographers, biologists, ecologists, physicist and modelers. I therefore recommend this article for publication. Nevertheless, I have some comments that may help to improve the manuscript before publication.

1. Given the lack of ENSO records, I can understand that the authors have chosen to cite Li et al. 2013, even though the latter reconstruction might have some limits - like any existing ENSO records - and therefore might not strictly reflect the past ENSO variability. Have the co-authors ever considered to compare their records with those for El Nino or La Nina generated by the NOAA since 1870 (https://psl.noaa.gov/enso/dashboard.html)? When looking at these latter records, it seems to me that there might be a better correspondence with their IPSO25 record than discussed in the manuscript. Could it change their interpretation regarding the impact of ENSO on the regional sea ice evolution if they would consider such records?

2. If I am not wrong, there is no clear statement on why the authors use both the TEX86-OH and TEX86-L. They should include few sentences explaining the differences between the two SST-derived proxies so that the non-experts would better understand what these two proxies mean and why they might show different patterns.

3. Simulations still hardly reproduce sea ice dynamic around Antarctica. This might be even more true in the Antarctic Peninsula given the strong seasonal contrast. I would therefore clearly highlight here the limitations of the model used on its representation of sea ice.

[Figure]

4. It is a bit disturbing to read section 4.1. before 4.2., 4.3. and 4.4 in a way that the authors use their proxies to reconstruct the last 200 years evolution of the oceanographic and sea ice conditions and then make the comparison with the model simulations, observations and ice core data. I am still wondering if it would not be more coherent and logical to first discuss the comparisons between their proxy records and available data and afterward propose hypotheses on sea ice variability for the last 200 years.

5. I am not surprised that their proxy records do not show a strong coherency with ice core and model data because (1) the ice core is located in James Ross Island, i.e. in the Northeastern Antarctica Peninsula, influenced by the Weddell gyre, where sea ice presence is almost year-round and therefore show a different climatic pattern than the one on the western side; and (2), models are still quite limited in reproducing properly sea ice cover. After carefully reading their conclusions, it sounds like the authors might not be so confident when interpretating their own data while they fit quite well with the satellite ones. I would suggest the authors to believe more in their data, bring forward the main issues with both the ice core and model estimations and posit that more data are needed in their studied area, especially on reconstructing air temperatures.

6. The authors have unique records spanning both the preindustrial and industrial periods, a transition during which there is a major increase in GHG. Nevertheless, the authors never link changes in sea ice with increasing $CO_2$ emissions for instance. Could they more clearly state or better explain if changes in sea ice could be related to any anthropogenic forcing? That would really interesting.

7. I do not see the need to show the campesterol, desmosterol or the B-sitosterol concentrations in the supplementary if they are not discussed in the main manuscript. I would suggest the authors to focus only on marine proxies helping to track sea ice dynamic and remove these records.

8. Although the manuscript already includes a lot of references, I would add a couple more. For instance, I would add two references on the modern local hydrography:

Dotto et al. (Multidecadal freshening and lightening in the deep waters of the Brasnfield Strait, Antarctica, JGR, 2016) and Ruiz Barlett et al. (On the temporal variability of intermediate and deep waters in the Western Basin of the Bransfield Strait, Deep-Sea Res., 2017). I would also add some on the modern sedimentation, Palanques et al. (Annual evolution of downward particle fluxes in the Western Bransfield Strait (Antarctica) during the FRUELA project, Deep-Sea Res. 2002), and the nutrient distribution and their influence on local marine productivity, Frants et al. (optimal multiparameter analysis of source water distributions in the Southern Drake Passage, Deep-Sea Res. 2013).

8. In the figure 1 captions, Abram et al. 2010 is mentioned 7 times which is a bit too much.

---

## Referee Comment (RC2) · Anonymous Referee #2 · 10 Sep 2020

SUMMARY COMMENTS

In this paper, Vorrath et al. present 3 new sea ice and temperature records of the past ∼200 years from a region of Antarctica that is known to have undergone rapid warming over recent decades. The manuscript presents new data, relevant discussion and is likely to attract a broad readership. However, the manuscript would be greatly improved if the authors used the post-1980 intervals of their cores to test/establish which aspects of the sea ice environment and/or ocean conditions are most likely driving the variations in their biomarker proxies before considering the pre-1980 record. The authors acknowledge the complexity of the sea ice environment but should do a more thorough

comparison of their youngest sediments with observations (and/or reanalysis data) to improve understanding of these dynamic settings and establish how best to interpret these proxy records for these specific locations before evaluating the pre-1980 records. The fact that the sediment cores have similar sedimentation rates, are close enough together to share large scale climatic/oceanographic regimes but distal enough to have site specific sea-ice conditions and water mass properties, makes these data ideal to investigate the more nuanced controls on production of the HBI and GDGT proxies.

MAJOR POINTS

> Model Results: Generally, Antarctic sea ice is poorly resolved in climate models so it is understandable that your model doesn't match with observations either. This mis-match between observations and model outputs throws serious doubt over the use of model results, especially for the pre-satellite era. Consider whether other lines of evidence are available (eg diatom concentrations or assemblage changes) that could support your interpretations. The general deficiency of models with regards Antarctic sea ice should be more broadly discussed in your manuscript, especially if you retain the model sea ice thickness results or any model results in your pre-satellite era records (eg. sea ice edge).

> Units: Not clear what the units are based on. If the 'units' reflect the numerical results then lines 422-423 should be moved to the end of the results section. Why have you not done any statistical analyses to determine the units, assess variability and/or sig-nificance of signals? If there is a reason, you should include it with lines 422-423. If the units are supposed to highlight the environmental interpretation then consider defining units for each site independently and then comparing the sequence/timing of units be-tween sites. The environmental summaries of the units are ambiguous and not very easy to follow with some of the descriptions seeming confused or even contradictory. eg. In unit B, 'moderate' spring sea ice is equated to 'long persisting' spring sea ice cover. Unit A described as - 'decreasing winter sea ice and moderate spring sea ice with low variability in seasonal sea ice changes.' So not clear how decreasing winter

sea ice can be reconciled persistent spring sea ice cover.

> Climate Links: I would encourage the authors to broaden their consideration of climate forcings/impacts to include eg. GHGs, the ozone hole, glacial meltwater/ice shelf collapse. They should also keep in mind that many of the climate modes/features are highly seasonal such that annual records may dilute the amplitude of a signal and produce weaker correlations.

MINOR POINTS

> Not always evident which results are used to inform interpretations. Connections between data and specific environmental conditions are clearly stated in some places but only sparsely, this should be improved throughout the manuscript but especially in the first section of the discussion.

> Where possible, replace neutral terms 'impact, influence, change' – with 'augment, elevate, increase, contribute to. . . etc.' or 'decrease, reduce, alleviate, mitigate . . .etc.' so the nature of feedbacks/responses are clear. Similarly, be sure that the nature of relationships/feedbacks are evident when 'impacts' are described as positive or negative.

> Replace 'at the WAP' with "in, on, along or through' the WAP' in your introduction. In your discussion, because your sites are only in one area of the WAP, you should replace WAP with Bransfield region or similar.

> Some of the refs could be updated with more recent literature eg. consider Montes-Hugo et al. (2009); Holland et al. (2012); Abernathey et al. (2016) & Frew et al. (2019).

Please also note the supplement to this comment:
https://cp.copernicus.org/preprints/cp-2020-63/cp-2020-63-RC2-supplement.pdf

**Supplement:**

**LINE SPECIFIC COMMENTS AND AMENDMENTS**

Not obvious what time frame you mean my 'since industrialisation' - this could be anything from 1750 to 1900... suggest you replace with clear time frame - either 'since specific year' OR 'over the last specific number decades' OR similar

45-53 Need to include timeframes for each of these 'observations' eg. rate of warming in AP only valid to a specific interval – not indefinitely.

'seems to be interrupted' understates the scale of decline recorded since 2014.

71-72 Phytoplankton are the source of dimethylsulphide not sea ice.

Replace 'impact' to 'contribute to'

Add 'climate' before 'mode'

84-86 Add reference.

Replace 'westerlies' with 'westerly winds'

Change describe to 'assert' or 'propose' or similar.

Change 'among' to 'between'

& 121 Explain why the IPSO signal is associated specifically with spring sea ice

123-124 Explain the relevance of analysing cores from different depths.

Add 'and SSI' after AP

Reverse order of 'yet not'

Replace Antarctic with 'AP'

Currents and water masses are not the same – ACC is a current; BSW is a water mass. Water masses can be carried/transported by currents but currents cannot produce water masses.

Replace Antarctic with 'AP'

159-161 Add that the areas of high PP are associated with diatom(?) production

Add 'Varied/heterogeneous' or similar before 'upwelling'

168-169 Delete 'input' and change 'silt and clay' to 'silts and clays'

Unless you used more than one multicorer you should change multicorers to 'a multicorer'

Need to include description of how 072-1 and 072-2 were correlated to match sample depths/ages.

Replace with 'Immediately after recovery, sediment cores were sectioned into 1 cm slices and subsampled. Samples designated for biomarker analyses were stored in glass vials at -XX deg C and samples for micropaleontological …'

Add 'was measured' or similar after either '226Ra' or 'depth interval'

201-202 Add info on sample size

Replace 'extracted' with 'processed'

Add 'of the residues/sample/' after fraction

Reword: not clear whether 'followed by temperature increase to 150 degC over 6 min, and to 325 degC within 57 min' is correct – do you mean 'the temperature was gradually increased to 150(325) deg C over the course of 6(57) min' or do you mean 'the temperature was raised to 150(325) deg C and sustained for 6(57) min.'

200-268 Clarify whether standards are used for the GDGT analyses

Need to state how many and the resolution of diatom samples as it is clearly different to the 1 cm resolution you describe in the section on sampling.

Slide preparation not described in Cardenas et al (2019), instead refers to method of Gersonde & Zeilinski, (2000)

Add justification for applying this TF for sea ice in the AP. Eg. how many of the 274 reference samples are from the AP?

Add justification for applying this TF for SST in the AP? How many of the 336 reference samples are from the AP?

Add 'proxies' after 'biogeochemical'

What about ice sheet dynamics?

Convert resolution of atmospheric model (1.9 degree) into km so it can be compared directly with ocean model.

Why is the resolution over the Arctic relevant here?

307-308 Reference/Evidence required. What about glacial isostatic adjustments?

– Is there a specific reason for using Dyer Plateau MSA record? Why not Bruce Plateau which is closer to the Bransfield Strait. Bruce Plateau record shows strong relationship between accumulation and sea ice cover too (Porter et al., 2016) but also has MSA record.

Delete 'coastal'. The Dyer Plateau site is on the spine of the Peninsula at >1900m elevation.

Give value of high F. curta contribution. Also need to include F. curta data in figure OR supplementary info. Consider including the diatom concentrations as one of the figures or supplementary info as another proxy for export productivity.

387- Since you provide an explanation for the outlier temperature in the SOTTEX record (392-393) you should also consider the strong divergence of trends between the biomarkers and WSI records, and the elevated SSST in the youngest sample.

Format 25 as subscript

392-393 Given that the SOTTEX exceeds realistic temperature min at the Trinity Island and Orleans Trough – can the other temperature points be trusted?

394-395 Either delete the statement 'A general weak cooling trend is present in SSST and SOTOH from 1920 CE to the 1990s.' or add an explanation and/or discussion point.

395-398 What is the evidence for increased mixing? More mixing would imply a convergence of both SOT temperature records and SST which is not evident in the SOTOH results from the Orleans Trough site.

'SSST corresponds quite well with SOTOH at PS97/056-1 and with SOTTEX at PS97/072-2'. What is the relevance/significance of this observation?

You need to give the values for the modern temperature range for the different areas when you make this type of assertion.

'…no clear dominance from one or the other regime is evident and we suggest that GDGT-derived temperatures are affected by influences of both BWS and WSW.' Do you mean both GDGT temperatures? How does this account for the distinct OH and TEX SOT records from the Orleans Trough site? You should also address why the temperatures derived from the SOTTEX are mostly warmer than the SOTOH temperatures.

As the site is farthest from the coast, please suggest the source(s) of this iron fertilisation

Replace 'have resulted from' with 'suggests' or 'indicates' or similar

Replace 'water' with 'subsurface ocean temperature(s)'

Delete either 'do not show such' or 'are not present'

Supplementary Table 7 not available (only supplementary Figure 1-5 were included in supplementary file).

Either delete 'coastal' or replace with offshore or similar

Does the 30x30 km model resolution apply to both atmosphere and ocean or just ocean? If both, this should be made clear in the methods section.

Insert 'in (the) model(s)' after 'missing feedbacks'. You should also state what the 'internal variabilities and missing feedbacks' are.

Replace 'recorded period' with the specific age range(s)

Consider changing 'be more related' to 'match' or 'more closely match' or 'reflect' or 'better reflect'

515-518 Should acknowledge that the proxy records do not resolve the observed patterns but that this could be a limitation of the resolution.

Have you considered using diatom concentrations, other diatom assemblage information and/or to convert your biomarker records into fluxes?

Why are the Bruce Plateau snow accumulation and MSA records not used? Bruce Plateau is much closer to your sites than Dyer Plateau.

It is not clear how your records are 'strongly influenced by the AP as a geographic barrier'. Please state what this influence is.

Please replace 'covered' with 'reflected' or 'represented' or similar

Consider replacing 'account for' with 'resolve' or 'capture'

This is the first mention of 'seasonal input of drift ice from the Weddell Sea'. You should include this in section 2.1 and state which of the sites are impacted by this and how.

Consider comment for lines 395-398 re: mixing at the Orleans Trough site.

Change 'picture' for 'depict'

589-590 If you are implying that the SOT temperatures are an annual signal you should state this clearly either in the paragraph at 247-250 or in your discussion of the temperature records.

Initials missing from ref.

**FIGURES**

> Figure 1 - In Moffat & Meredith (2018) West of Anvers Island the APCC is illustrated as flowing southwards. Please check.

Delete replicates of 'Abram et al 2010' in figure caption.

Intervals when the Orleans Trough site has SOTtex 2-3 degC warmer than the Trinity Island and East Bransfield Basin site. Please provide some potential explanations for these offsets in the discussion.

> Figure 2 - Reword: 'extrapolating ages based on sedimentation rate of oldest 3 cm' – I think you mean the 3 cm of sediments at the base of the 'reliably' dated interval not the oldest 3 cm.

Add comments on the changes in sedimentation rates to your discussion.

> Figure 4 - Add sample markers to biomarker data to show sample numbers/intervals.

Add comments/discussion on why transfer function WSI % is generally much higher than observed.

> Figure 6: - Include the modelled SSST to compare with diatom-derived SSST.

SUGGESTED REFS

> Abernathey et al. (2016) Water-mass transformation by sea ice in the upper branch of the Southern Ocean overturning. *Nature Geoscience*, 9. 8 pp. doi:10.1038/ngeo2749

> Frew et al. (2019) Sea ice – ocean feedbacks in the Antarctic shelf seas. *Journal of Physical Oceanography*, 49. 2423-2446. doi:10.1175/JPO-D-18-0229.1

> Holland et al. (2012) Wind-driven trends in Antarctic sea-ice drift. *Nature Geoscience*, 5. 872-875. doi:10.1038/ngeo1627

> Montes-Hugo et al. (2009) Recent Changes in Phytoplankton Communities Associated with Rapid Regional Climate Change Along the Western Antarctic Peninsula. *Science*, doi:10.1126/science.1164533

> Porter et al. (2016) Bellingshausen Sea ice extent recorded in an Antarctic Peninsula ice core, *Journal of Geophysical Research Atmosphere*, 121, doi:10.1002/2016JD025626.

---

## Author Comment (AC1) · 9 Oct 2020

Dear Editor, dear Reviewers 1 and 2,

Below please find our response to your comments. We appreciate your helpful and constructive comments and suggestions, which have helped to improve many sections of our manuscript. Especially the suggested references were very useful and gave valuable input to the discussion. We revised our manuscript and will upload the revised version as soon as possible. We added a new figure (Fig. 8 in the manuscript, see attached supplement 1 to our response) that summarizes the different sea ice conditions along the Bransfield Strait. We added Gerhard Kuhn as a co-author because of his data contribution for the sediment dating process.

Response to comments of Reviewer 1: Given the lack of ENSO records, I can understand that the authors have chosen to cite Li et al. 2013, even though the latter reconstruction might have some limits - like any existing ENSO records - and therefore might not strictly reflect the past ENSO variability. Have the co-authors ever considered to compare their records with those for El Nino or La Nina generated by the NOAA since 1870 (https://psl.noaa.gov/enso/dashboard.html)? When looking at these latter records, it seems to me that there might be a better correspondence with their IPSO25 record than discussed in the manuscript. Could it change their interpretation regarding the impact of ENSO on the regional sea ice evolution if they would consider such records?

RESPONSE: We took the NOAA ENSO data you recommended to us and correlated it to our sea ice proxy (Pearson's correlation test in R, package Performance Analytics). Although the visual correspondence seems to be higher, the statistical correlation was almost as low as with the ENSO data from Li et al. (2013)(see attached supplement 2 to our response). We decided to keep the ENSO data from Li et al. (2013) since they cover a longer time series reaching back to 1800.

If I am not wrong, there is no clear statement on why the authors use both the TEX86-OH and TEX86-L. They should include few sentences explaining the differences between the two SST-derived proxies so that the non-experts would better understand what these two proxies mean and why they might show different patterns.

RESPONSE: The application of both hydroxylated and non-hydroxylated GDGT temperature estimations follows recommendation by Fietz et al. (2020). Their significance for different ocean regions is still a subject under discussion, especially for low temperature paleo events at continental margins. We now explain our decision to use both TEX86-OH and TEX86-L in the revised methods section.

Simulations still hardly reproduce sea ice dynamic around Antarctica. This might be even more true in the Antarctic Peninsula given the strong seasonal contrast. I would therefore clearly highlight here the limitations of the model used on its representation of sea ice.

RESPONSE: We addressed the limitation of the modelled data in the revised discussion (section 4.2) with "While modelled and satellite derived data have similar ocean grid sizes (model: 30x30 km, satellite: 25x25 km) we suppose that global models such as AWI-ESM2 cannot resolve the AP sub-aerial and marine topography and have difficulties in capturing local to regional near coastal sea-ice dynamics in the study region. Another reason is related to internal variability and missing feedbacks in the model which makes a direct comparison of short time series difficult. Changes in the forcings are restricted to the insolation and greenhouse gases and can affect the simulated climate by bringing in natural noises. For the 240 years of modelled period, especially during small changes, the internal variabilities can dominate the climate change bringing difficulties to model-data comparisons. Feedbacks of aerosols, ozone, ice sheet dynamics, dust, solar and volcano activity are missing because these elements were static. Further, the modelled Antarctic sea ice is generally thicker and the coverage is higher due to a reduced warming of the Southern Ocean within the model setup (Sidorenko et al., 2019)."

It is a bit disturbing to read section 4.1. before 4.2., 4.3. and 4.4 in a way that the authors use their proxies to reconstruct the last 200 years evolution of the oceanographic and sea ice conditions and then make the comparison with the model simulations, observations and ice core data. I am still wondering if it would not be more coherent and logical to first discuss the comparisons between their proxy records and available data and afterward propose hypotheses on sea ice variability for the last 200 years.

RESPONSE: As suggested, we changed the order of the discussion and shifted the stratigraphic interpretation to section 4.6. We renamed this part with "Interpretation of combined paleoenvironmental biomarkers". We also revised the stratigraphic interpretation and defined environmental units and removed the unit bar from figures 5-7.

I am not surprised that their proxy records do not show a strong coherency with ice core and model data because (1) the ice core is located in James Ross Island, i.e. in the Northeastern Antarctica Peninsula, influenced by the Weddell gyre, where sea ice presence is almost year-round and therefore show a different climatic pattern than the one on the western side; and (2), models are still quite limited in reproducing properly sea ice cover. After carefully reading their conclusions, it sounds like the authors might not be so confident when interpretating their own data while they fit quite well with the satellite ones. I would suggest the authors to believe more in their data, bring forward the main issues with both the ice core and model estimations and posit that more data are needed in their studied area, especially on reconstructing air temperatures.

RESPONSE: Thank you for your comment. We now address the issue with the geographical position of the ice core record from JRI and the limitations of the model when comparing with our biomarker data (in section 4.3) as following: "We relate the divergence of ice core and modelled sea ice data from our sediment core data firstly, to the different spatial coverage and geographic origin of the environmental signals archived within the ice cores and, secondly, the aspect that AWI-ESM2 cannot resolve the AP sub-aerial and marine topography and have difficulties in capturing local to regional near coastal sea-ice dynamics in the study region. clearly shows that available sea ice reconstructions still do not reflect sea ice properties in the Bransfield Strait In fact, our sediment records reflect local to regional impact of the BSW and WSW that carry opposite sea ice and water mass properties and neither represent sea ice properties of the Bellingshausen Sea nor the Weddell Sea. In addition, as the AP is acting as a geographic barrier between these water masses, the region is highly sensitive to oceanographic variabilities driven by atmospheric patterns. For example, it is suggested that strong westerly winds and a positive SAM diminish the inflow of WSW into the Bransfield Strait (Dotto et al., 2016)."

The authors have unique records spanning both the preindustrial and industrial periods, a transition during which there is a major increase in GHG. Nevertheless, the authors never link changes in sea ice with increasing CO2 emissions for instance. Could they more clearly state or better explain if changes in sea ice could be related to any anthropogenic forcing? That would really interesting.

RESPONSE: As rising greenhouse gases increase atmospheric and ocean temperature we included your suggestion in section 4.3 and added local atmospheric temperature data as well as CO2 concentrations in Figure 6 of the main text.

I do not see the need to show the campesterol, desmosterol or the B-sitosterol concentrations in the supplementary if they are not discussed in the main manuscript. I would suggest the authors to focus only on marine proxies helping to track sea ice dynamic and remove these records.

RESPONSE: We agree with your comment and have removed campesterol, desmosterol and beta-sitosterol from the supplementary information. The data can still be found in the supplementary tables on https://doi.pangaea.de/10.1594/PANGAEA.918732

Although the manuscript already includes a lot of references, I would add a couple more. For instance, I would add two references on the modern local hydrography: Dotto et al. (Multidecadal freshening and lightening in the deep waters of the Bransfield Strait, Antarctica, JGR, 2016) and Ruiz Barlett et al. (On the temporal variability of intermediate and deep waters in the Western Basin of the Bransfield Strait, Deep-Sea Res., 2017). I would also add some on the modern sedimentation, Palanques et al. (Annual evolution of downward particle fluxes in the Western Bransfield Strait (Antarctica) during the FRUELA project, Deep-Sea Res. 2002), and the nutrient distribution and their influence on local marine productivity, Frants et al. (optimal multiparameter analysis of source water distributions in the Southern Drake Passage, Deep-Sea Res. 2013).

RESPONSE: Thank you very much for the very helpful literature suggestions. Especially Dotto et al. (2016) and Barlett et al. (2017) have been very helpful in improving the discussion in section 4.5 of our manuscript.

In the figure 1 captions, Abram et al. 2010 is mentioned 7 times which is a bit too much.

RESPONSE: This has been corrected.
* * *
Response to comments of Reviewers 2: In this paper, Vorrath et al. present 3 new sea ice and temperature records of the past âĹij200 years from a region of Antarctica that is known to have undergone rapid warming over recent decades. The manuscript presents new data, relevant discussion and is likely to attract a broad readership. However, the manuscript would be greatly improved if the authors used the post-1980 intervals of their cores to test/establish which aspects of the sea ice environment and/or ocean conditions are most likely driving the variations in their biomarker proxies before considering the pre-1980 record. The authors acknowledge the complexity of the sea ice environment but should do a more thorough comparison of their youngest sediments with observations (and/or reanalysis data) to improve understanding of these dynamic settings and establish how best to interpret these proxy records for these specific locations before evaluating the pre-1980 records. The fact that the sediment cores have similar sedimentation rates, are close enough together to share large scale climatic/oceanographic regimes but distal enough to have site specific sea-ice conditions and water mass properties, makes these data ideal to investigate the more nuanced controls on production of the HBI and GDGT proxies.

MAJOR POINTS > Model Results: Generally, Antarctic sea ice is poorly resolved in climate models so it is understandable that your model doesn't match with observations either. This mismatch between observations and model outputs throws serious doubt over the use of model results, especially for the pre-satellite era. Consider whether other lines of evidence are available (eg diatom concentrations or assemblage changes) that could support your interpretations. The general deficiency of models with regards Antarctic sea ice should be more broadly discussed in your manuscript, especially if you retain the model sea ice thickness results or any model results in your pre-satellite era records (eg. sea ice edge).

RESPONSE: Following your comment we included Fragilariopsis curta (a characteristic diatom species associated with sea ice) in our records and show it in Fig. 3-5. We now address the limitations of the model in our discussion (section 4.2 and 4.3; please see response to reviewer 1 above).

> Units: Not clear what the units are based on. If the 'units' reflect the numerical results then lines 422-423 should be moved to the end of the results section. Why have you not done any statistical analyses to determine the units, assess variability and/or significance of signals? If there is a reason, you should include it with lines 422-423. If the units are supposed to highlight the environmental interpretation then consider defining units for each site independently and then comparing the sequence/timing of units between sites. The environmental summaries of the units are ambiguous and not very easy to follow with some of the descriptions seeming confused or even contradictory. eg. In unit B, 'moderate' spring sea ice is equated to 'long persisting' spring sea ice cover. Unit A described as - 'decreasing winter sea ice and moderate spring sea ice with low variability in seasonal sea ice changes.' So not clear how decreasing winter sea ice can be reconciled persistent spring sea ice cover.

RESPONSE: We revised our stratigraphic interpretation and defined environmental units instead of stratigraphic units depending on our sea ice proxies (now section 4.6) and removed the unit bars from figures 5-7. We did not use statistical analyses because of the low resolution of our records.

> Climate Links: I would encourage the authors to broaden their consideration of climate forcings/impacts to include eg. GHGs, the ozone hole, glacial meltwater/ice shelf collapse. They should also keep in mind that many of the climate modes/features are highly seasonal such that annual records may dilute the amplitude of a signal and produce weaker correlations.

RESPONSE: We added the concentration of atmospheric CO2 (Fig. 6) and further discussed glacial meltwater input and the contribution of Weddell Sea Water related to the strength of the Southern Westerly Winds (section 4.5). We agree with the reviewer that highly seasonal climate modes could be overseen in our records that reflect spring to summer (and potentially annual) environmental conditions which may result in weaker correlations. The atmospheric patterns ENSO and SAM have a strong impact on the spring season so we clarified in section 4.5 "While sediment records integrate environmental conditions of several years, the influence of highly seasonal climate modes or events may not be properly dissolved which could explain the weaker relationships. However, since atmospheric circulation affects the heat and sea ice distribution along the WAP especially during spring time (Clem et al., 2016), we expect patterns of ENSO and/or SAM to leave a footprint in our spring sea ice IPSO25 record."

MINOR POINTS > Not always evident which results are used to inform interpretations. Connections between data and specific environmental conditions are clearly stated in some places but only sparsely, this should be improved throughout the manuscript but especially in the first section of the discussion.

RESPONSE: We revised the manuscript considering this comment. Especially in section 4.1 and 4.6 we connected the given data to their environmental interpretations. For example, in section 4.1 "The remarkably low SOTTEX in the year 2006 CE might be a result of cold meltwater injections due to enhanced glacier melting (e.g. Pastra Glacier on Trinity Island). At the same time high SSST and SOTOH as well as a low WSI point towards a significantly warm period around the year 2006 which is underlined by meteorological station data (Turner et al., 2019)."

> Where possible, replace neutral terms 'impact, influence, change' – with 'augment, elevate, increase, contribute to. . . etc.' or 'decrease, reduce, alleviate, mitigate . . .etc.' so the nature of feedbacks/responses are clear. Similarly, be sure that the nature of relationships/feedbacks are evident when 'impacts' are described as positive or negative.

RESPONSE: We carefully revised our manuscript and improved the wording.

> Replace 'at the WAP' with "in, on, along or through' the WAP' in your introduction. In your discussion, because your sites are only in one area of the WAP, you should replace WAP with Bransfield region or similar.

RESPONSE: We revised this in the introduction (mostly using "along the WAP") and carefully considered your suggestion for the discussion by calling the study site the Bransfield Strait (and the specific core sites) and not the WAP.

> Some of the refs could be updated with more recent literature eg. consider Montes-Hugo et al. (2009); Holland et al. (2012); Abernathey et al. (2016) & Frew et al. (2019).

RESPONSE: Thank you very much. We considered theses references in the discussion and included them in the reference list.

Please also note the supplement to this comment: https://cp.copernicus.org/preprints/cp-2020-63/cp-2020-63-RC2-supplement.pdf

LINE SPECIFIC COMMENTS AND AMENDMENTS 44 Not obvious what time frame you mean my 'since industrialisation' - this could be anything from 1750 to 1900... suggest you replace with clear time frame - either 'since specific year' OR 'over the last specific number decades' OR similar

RESPONSE: As 'industrialisation' is defined so differently by many sources we preferred to change our title and use the term "past 200 years" instead. We further use the term 'industrialisation' only in context with clear definitions by e.g. IPCC references or in connection with the model setup (in section 2.5).

45-53 Need to include timeframes for each of these 'observations' eg. rate of warming in AP only valid to a specific interval – not indefinitely.

RESPONSE: We added the specific time intervals for each observation.

seems to be interrupted' understates the scale of decline recorded since 2014.

RESPONSE: We corrected this.

71-72 phytoplankton are the source of dimethylsulphide not sea ice.

RESPONSE: We corrected this.

Replace 'impact' to 'contribute to'

RESPONSE: We corrected this.

Add 'climate' before 'mode'

RESPONSE: We corrected this.

84-86 Add reference.

RESPONSE: We added Stammerjohn et al. 2008.

Replace 'westerlies' with 'westerly winds'

RESPONSE: We corrected this and used "westerly winds" in connection with several literature sources and "westerly wind belt" when referring to Koffmann et al. (2014).

Change describe to 'assert' or 'propose' or similar.

RESPONSE: We changed this.

Change 'among' to 'between'

RESPONSE: We changed this.

& 121 Explain why the IPSO signal is associated specifically with spring sea ice

RESPONSE: During spring, increased light availability and ice melt permit the growth of sea ice algae (including B. adeliensis as the main source organism for IPSO25) and it is thus assumed that IPSO25 (and its Arctic Ocean counterpart IP25) reflect spring sea ice conditions. In the text, we now also refer to Riaux-Gobin and Poulin (2004) and Belt et al. (2016).

123-124 Explain the relevance of analysing cores from different depths.

RESPONSE: We added that the three core sites at different depths were used because they cover different parts (i.e. oceanic settings) of the Bransfield Region.

Add 'and SSI' after AP

RESPONSE: We corrected this.

Reverse order of 'yet not'

RESPONSE: We corrected this.

Replace Antarctic with 'AP'

RESPONSE: We corrected this.

Currents and water masses are not the same – ACC is a current; BSW is a water mass. Water masses can be carried/transported by currents but currents cannot produce water masses.

RESPONSE: We corrected the wording regarding currents and water masses.

Replace Antarctic with 'AP'

RESPONSE: We corrected this.

159-161 Add that the areas of high PP are associated with diatom(?) production

RESPONSE: We added this.

Add 'Varied/heterogeneous' or similar before 'upwelling'

RESPONSE: We added "heterogeneous".

168-169 Delete 'input' and change 'silt and clay' to 'silts and clays'

RESPONSE: We corrected this.

Unless you used more than one multicorer you should change multicorers to 'a multicorer'

RESPONSE: We corrected this.

Need to include description of how 072-1 and 072-2 were correlated to match sample depths/ages.

RESPONSE: We added the correlation of both cores in the supplements (supplement S1) and referred to it in the text. The TOC content of both cores was measured and compared to each other to apply the age model of 072-2 on 071-1.

Replace with 'Immediately after recovery, sediment cores were sectioned into 1 cm slices and subsampled. Samples designated for biomarker analyses were stored in glass vials at -XX deg C and samples for micropaleontological ...'

RESPONSE: We replaced it.

Add 'was measured' or similar after either '226Ra' or 'depth interval'

RESPONSE: We added this.

201-202 Add info on sample size

RESPONSE: We added the sample size.

Replace 'extracted' with 'processed'

RESPONSE: We replaced it.

Add 'of the residues/sample/' after fraction

RESPONSE: We added this.

Reword: not clear whether 'followed by temperature increase to 150 degC over 6 min, and to 325 degC within 57 min' is correct – do you mean 'the temperature was gradually increased to 150(325) deg C over the course of 6(57) min' or do you mean

'the temperature was raised to 150(325) deg C and sustained for 6(57) min.'

RESPONSE: We refer to a gradual increase of temperature and reworded this part.

200-268 Clarify whether standards are used for the GDGT analyses

RESPONSE: We added information about all standards in section 2.3.

Need to state how many and the resolution of diatom samples as it is clearly different to the 1 cm resolution you describe in the section on sampling.

RESPONSE: We added this information. We studied every second centimeter from core 056-1 and every centimeter from cores 068-2 and 072-1 (see section 2.4).

Slide preparation not described in Cardenas et al (2019), instead refers to method of Gersonde & Zeilinski, (2000)

RESPONSE: We corrected this.

Add justification for applying this TF for sea ice in the AP. Eg. how many of the 274 reference samples are from the AP?

RESPONSE: We justified the TF with the circum-Antarctic data coverage and added the reference samples from the AP.

Add justification for applying this TF for SST in the AP? How many of the 336 reference samples are from the AP?

RESPONSE: See above.

Add 'proxies' after 'biogeochemical'

RESPONSE: We added this.

What about ice sheet dynamics?

RESPONSE: We added the information about how the model addresses different variables (including ice sheet dynamics) in the text (section 4.2): "Feedbacks of aerosols, ozone, ice sheet dynamics, dust, solar and volcano activity are missing because these elements were static."

Convert resolution of atmospheric model (1.9 degree) into km so it can be compared directly with ocean model.

RESPONSE: We added the km.

Why is the resolution over the Arctic relevant here?

RESPONSE: The resolution is highest in both polar areas, Arctic and Antarctic, we added this to the text (section 2.5).

307-308 Reference/Evidence required. What about glacial isostatic adjustments?

RESPONSE: We added the following information: "We used a fixed ice sheet and the topography was taken from ICE6G (Peltier et al., 2015) so isostatic adjustments are neglected."

– Is there a specific reason for using Dyer Plateau MSA record? Why not Bruce Plateau which is closer to the Bransfield Strait. Bruce Plateau record shows strong relationship between accumulation and sea ice cover too (Porter et al., 2016) but also has MSA record.

RESPONSE: We took the MSA record from Dyer Plateau but also the stacked record (including records from James Ross Island and Beethoven Peninsula, Abram et al., 2010) because we wanted to show a record that represents the sea ice development in the Bellingshausen quite well over the longest available period (last 120 years). We included the snow accumulation data from Bruce Plateau, while Bruce Plateau MSA data is not publicly available.

Delete 'coastal'. The Dyer Plateau site is on the spine of the Peninsula at >1900m elevation.

RESPONSE: We corrected this.

Give value of high F. curta contribution. Also need to include F. curta data in figure OR supplementary info. Consider including the diatom concentrations as one of the figures or supplementary info as another proxy for export productivity.

RESPONSE: We included F. curta in Fig. 3-5. Unfortunately, the complete diatom data set of one core was not available and we were not able to provide diatom concentrations.

387- Since you provide an explanation for the outlier temperature in the SOTTEX record (392-393) you should also consider the strong divergence of trends between the biomarkers and WSI records, and the elevated SSST in the youngest sample.

RESPONSE: We included this divergence and referred the low WSI and high SSST records to a very warm period that was also recorded in meteorological station data in 2006.

392-393 Given that the SOTTEX exceeds realistic temperature min at the Trinity Island and Orleans Trough – can the other temperature points be trusted?

RESPONSE: We addressed the uncertainties related to GDGT-derived temperatures for these cores in the methods section 3.2 and point out that SOT-TEX might not reflect exact/absolute temperatures but rather a temperature trend. We also referred to Fietz et al. (2020) who suggest OH-based temperature estimations to better suit polar regions than TEX-based estimations.

394-395 Either delete the statement 'A general weak cooling trend is present in SSST and SOTOH from 1920 CE to the 1990s.' or add an explanation and/or discussion point.

RESPONSE: We rephrased this.

395-398 What is the evidence for increased mixing? More mixing would imply a convergence of both SOT temperature records and SST which is not evident in the SOTOH results from the Orleans Trough site.

RESPONSE: We suggest increased mixing from the description of oceanography (Sangrà et al., 2011) and the position of the core. We discuss the contradiction of high SOT-TEX against low SOT-OH and SSST in section 4.1.

'SSST corresponds quite well with SOTOH at PS97/056-1 and with SOTTEX at PS97/072-2'. What is the relevance/significance of this observation?

RESPONSE: We thank the reviewer and agree that this observation is not relevant here. We deleted this part.

You need to give the values for the modern temperature range for the different areas when you make this type of assertion.

RESPONSE: After Cook et al. (2016) the modern temperature values in the Bransfield Strait are below -0.5°C (in the upper 400m). We added this information to the text.

'...no clear dominance from one or the other regime is evident and we suggest that GDGT-derived temperatures are affected by influences of both BWS and WSW.' Do you mean both GDGT temperatures? How does this account for the distinct OH and TEX SOT records from the Orleans Trough site? You should also address why the temperatures derived from the SOTTEX are mostly warmer than the SOTOH temperatures.

RESPONSE: We are sorry, this was a bit misleading. We were thinking that the exceptional high SOT-TEX would result from the interplay of warmer BWS and colder WSW but as this is not evident in the other temperature records, we dismissed this idea. We have to admit that we cannot answer why SOT-TEX is much higher than the other temperature records but following Fietz et al. (2020) it seems that SOT-TEX is less suitable for polar temperature estimation than SOT-OH. We keep the record of SOT-TEX in our data as it highlights a potential weakness of TEX-based temperature reconstructions in polar regions.

As the site is farthest from the coast, please suggest the source(s) of this iron

fertilization.

RESPONSE: We addressed this.

Replace 'have resulted from' with 'suggests' or 'indicates' or similar

RESPONSE: We replaced it.

Replace 'water' with 'subsurface ocean temperature(s)'

RESPONSE: We replaced it.

Delete either 'do not show such' or 'are not present'

RESPONSE: We corrected this.

Supplementary Table 7 not available (only supplementary Figure 1-5 were included in supplementary file).

RESPONSE: All tables will be available on Pangaea.de after publication (https://doi.pangaea.de/10.1594/PANGAEA.897165).

Either delete 'coastal' or replace with offshore or similar

RESPONSE: We deleted "coastal".

Does the 30x30 km model resolution apply to both atmosphere and ocean or just ocean? If both, this should be made clear in the methods section.

RESPONSE: We clarified that the resolution is only for the ocean with "While modelled and satellite derived data have similar ocean grid sizes (model: 30x30 km, satellite: 25x25 km) we suppose that global models such as AWI-ESM2 cannot resolve the AP sub-aerial and marine topography and have difficulties in capturing local to regional near coastal sea-ice dynamics in the study region." The atmosphere resolution is described in the methods as 1.9 degree (or 210 km) (section 2.5).

Insert 'in (the) model(s)' after 'missing feedbacks'. You should also state what the

'internal variabilities and missing feedbacks' are.

RESPONSE: We added further information about feedbacks and internal variabilities: "Changes in the forcings are restricted to the insolation and greenhouse gases and can affect the simulated climate by bringing in natural noises. For the 240 years of modelled period, especially during small changes, the internal variabilities can dominate the climate change bringing difficulties to model-data comparisons. Feedbacks of aerosols, ozone, ice sheet dynamics, dust, solar and volcano activity are missing because these elements were static."

Replace 'recorded period' with the specific age range(s)

RESPONSE: We corrected this.

Consider changing 'be more related' to 'match' or 'more closely match' or 'reflect' or 'better reflect'

RESPONSE: We use "better reflect".

515-518 Should acknowledge that the proxy records do not resolve the observed patterns but that this could be a limitation of the resolution.

RESPONSE: We considered the resolution and rephrased it.

Have you considered using diatom concentrations, other diatom assemblage information and/or to convert your biomarker records into fluxes?

RESPONSE: We revised the presentation of our data and included F. curta since this diatom shows the most significant changes regarding sea ice. Diatom concentrations were not available for core 068-2, so for a consistent discussion we did not take into account diatom concentrations or fluxes.

Why are the Bruce Plateau snow accumulation and MSA records not used? Bruce Plateau is much closer to your sites than Dyer Plateau.

RESPONSE: The MSA records from Bruce Plateau were not publicly available but we included the snow accumulation in our data and in the discussion.

It is not clear how your records are 'strongly influenced by the AP as a geographic barrier'. Please state what this influence is.

RESPONSE: We rephrased this part (section 4.3) and explain that the AP acts as a barrier between the different water masses of Bellingshausen Sea Water and Weddell Sea Water which, in turn, influences the oceanographic setting in the Bransfield Strait.

Please replace 'covered' with 'reflected' or 'represented' or similar

RESPONSE: We changed it.

Consider replacing 'account for' with 'resolve' or 'capture'

RESPONSE: We replaced it.

This is the first mention of 'seasonal input of drift ice from the Weddell Sea'. You should include this in section 2.1 and state which of the sites are impacted by this and how.

RESPONSE: We included this in section 2.1.

Consider comment for lines 395-398 re: mixing at the Orleans Trough site.

RESPONSE: We considered the comment and changed the text accordingly.

Change 'picture' for 'depict'

RESPONSE: We changed it.

589-590 If you are implying that the SOT temperatures are an annual signal you should state this clearly either in the paragraph at 247-250 or in your discussion of the temperature records.

RESPONSE: We assume that both GDGT-based temperatures estimations reflect annual mean temperatures because their calibrations are also based on annual mean SST (Kim et al., 2010; Lü et al., 2015).

Initials missing from ref.

RESPONSE: We added the initials.

FIGURES > Figure 1 - In Moffat & Meredith (2018) West of Anvers Island the APCC is illustrated as flowing southwards. Please check.

RESPONSE: The APCC flows southward south of Anvers Island. In the Gerlache Strait, between Anvers Island and Trinity Island, the direction of the APCC is northwards. We corrected this in the figure and in the study area description.

Delete replicates of 'Abram et al 2010' in figure caption.

RESPONSE: We are sorry. This is a bug from Mendeley. We corrected this.

Intervals when the Orleans Trough site has SOTtex 2-3 degC warmer than the Trinity Island and East Bransfield Basin site. Please provide some potential explanations for these offsets in the discussion.

RESPONSE: We discuss this temperature offset in section 4.1.

> Figure 2 - Reword: 'extrapolating ages based on sedimentation rate of oldest 3 cm' – I think you mean the 3 cm of sediments at the base of the 'reliably' dated interval not the oldest 3 cm.

RESPONSE: We reworded and corrected it.

Add comments on the changes in sedimentation rates to your discussion.

RESPONSE: We added comments on this issue in section 4.1.

> Figure 4 - Add sample markers to biomarker data to show sample numbers/intervals.

RESPONSE: We added the markers.

Add comments/discussion on why transfer function WSI % is generally much higher than observed.

RESPONSE: We added a comment in section 4.2 about this.

> Figure 6: - Include the modelled SSST to compare with diatom-derived SSST.

RESPONSE: As all modelled ocean temperatures are quite similar we chose to show the annual SOT only.

SUGGESTED REFS > Abernathey et al. (2016) Water-mass transformation by sea ice in the upper branch of the Southern Ocean overturning. Nature Geoscience, 9. 8 pp. doi:10.1038/ngeo2749 > Frew et al. (2019) Sea ice – ocean feedbacks in the Antarctic shelf seas. Journal of Physical Oceanography, 49. 2423-2446. doi:10.1175/JPO-D-18-0229.1 > Holland et al. (2012) Wind-driven trends in Antarctic sea-ice drift. Nature Geoscience, 5. 872-875. doi:10.1038/ngeo1627 > Montes-Hugo et al. (2009) Recent Changes in Phytoplankton Communities Associated with Rapid Regional Climate Change Along the Western Antarctic Peninsula. Science, doi:10.1126/science.1164533 > Porter et al. (2016) Bellingshausen Sea ice extent recorded in an Antarctic Peninsula ice core, Journal of Geophysical Research Atmosphere, 121, doi:10.1002/2016JD025626.

RESPONSE: We thank you for the references, which have been incorporated into the discussion and included in the reference list.

[Figure]

*Supplement 1 (Figure 8 in the manuscript): The illustration of the four environmental units and their dominant drivers of a) low sea ice cover and high ocean temperatures from dominating BSW described in Unit A, b) moderate winter and spring sea ice cover with decreasing temperatures that propagate from the Weddell Sea to the southern WAP in Unit B, c) high but variable sea ice cover and ocean temperature lows under advancing BSW and +SAM and additional meltwater pulses of Unit C, and d) high sea ice cover under a warm ocean temperature reversal as a result of BSW dominance and meltwater input from the AP in Unit D.*

**Fig. 1.** Supplement_1_Environmental Units

[Figure]

Supplement 2: The correlations of IPSO$_{25}$ values from core site 56, 68 and 72 with ENSO data from NOAA (ENSO3.4) and Li et al. (2013). Each significance level is associated to a symbol (p-values are 0.001 = "***"; 0.01 = "**"; 0.05 = "*"; 0.1 = "."; 0/1 = " ").

**Fig. 2.** Supplement_2_Correlations